# Pulcherriminic acid modulates iron availability and protects against oxidative stress during microbial interactions

Vincent Charron-Lamoureux [1] ✉, Lounès Haroune[2,3,4], Maude Pomerleau[1,4], Léo Hall[2,4], Frédéric Orban[2,4], Julie Leroux[1], Adrien Rizzi [2], Jean-Sébastien Bourassa[1], Nicolas Fontaine[2], Élodie V. d'Astous[2], Philippe Dauphin-Ducharme [2], Claude Y. Legault [2], Jean-Philippe Bellenger [2] & Pascale B. Beauregard [1] ✉

Siderophores are soluble or membrane-embedded molecules that bind the oxidized form of iron, Fe(III), and play roles in iron acquisition by microorganisms. Fe(III)-bound siderophores bind to specific receptors that allow microbes to acquire iron. However, certain soil microbes release a compound (pulcherriminic acid, PA) that, upon binding to Fe(III), forms a precipitate (pulcherrimin) that apparently functions by reducing iron availability rather than contributing to iron acquisition. Here, we use *Bacillus subtilis* (PA producer) and *Pseudomonas protegens* as a competition model to show that PA is involved in a peculiar iron-managing system. The presence of the competitor induces PA production, leading to precipitation of Fe(III) as pulcherrimin, which prevents oxidative stress in *B. subtilis* by restricting the Fenton reaction and deleterious ROS formation. In addition, *B. subtilis* uses its known siderophore bacillibactin to retrieve Fe(III) from pulcherrimin. Our findings indicate that PA plays multiple roles by modulating iron availability and conferring protection against oxidative stress during inter-species competition.

Iron (Fe) is an essential metal required in high cellular concentrations for the survival and growth of most living organisms on earth. While abundant in soil, its solubility and, hence its bioavailability are very low at circumneutral pH (ca $10^{-10}$ M)[1]. To solve this conundrum of high demand but low supply, soil bacteria produce siderophores–low molecular-weight compounds with high affinity for Fe(III)[2]. Specific transport systems allow Fe to be taken by cells and utilized in various cellular pathways and proteins[3–5]. Soil microorganisms often compete for Fe and have evolved mechanisms to maximize Fe acquisition, including the production of multiple siderophores with a range of affinities for Fe and the expression of various receptors at the cell surface capable of binding siderophores, sometimes including xenosiderophores[6–9]. In interspecies competition, secretion of siderophore can confer growth advantage through rapid Fe monopolization[10]. However, Fe sequestration by siderophores can be a double-edged sword as competitors can also cheat them, thus acquiring Fe without sharing the metabolic cost of siderophore biosynthesis[9,11].

Using the same synthesis pathway, *B. subtilis* produces two siderophores: the relatively weak binding-affinity 2,3-dihydroxybenzoic acid (DHB; one catechol function) and the strong binding-affinity bacillibactin (BB; three catechol function)[12]. Pulcherriminic acid (PA), another Fe-chelating molecule produced by yeast and many bacteria including *B. subtilis*[13–15], harbors two hydroxamate groups and forms

[1]Département de biologie, Faculté des sciences, Université de Sherbrooke, Sherbrooke, QC, Canada. [2]Département de chimie, Faculté des sciences, Université de Sherbrooke, Sherbrooke, QC, Canada. [3]Institut de pharmacologie de Sherbrooke, Faculté de médecine, Université de Sherbrooke, Sherbrooke, QC, Canada. [4]These authors contributed equally: Lounès Haroune, Maude Pomerleau, Léo Hall, Frédéric Orban. ✉e-mail: Vincent.Charron-Lamoureux@USherbrooke.ca; Pascale.B.Beauregard@USherbrooke.ca

the red insoluble pulcherrimin complex upon binding Fe(III)[15–17]. Recently, the extracellular accumulation of pulcherrimin (PA-Fe) was shown to restrict biofilm expansion, and it was suggested that PA act as an intercellular signal triggering the transition from exponential to stationary phase in *B. subtilis*[14,18]. Pulcherrimin was first described in a small group of yeasts, including *Candida* spp. and *Kluyveromyces* spp.[15,19]. In these organisms, PA production was shown to have antagonistic activity, probably through Fe depletion[20]. In *Kluyveromyces lactis*, it was suggested that PA might act as a siderophore, a contradictory idea since the pulcherrimin complex is insoluble and, apparently, not a bioavailable source of Fe[19,21]. Indeed, nearly all previous studies in yeast and *Bacillus* stipulate that PA irreversibly binds Fe(III) and thus plays no role in Fe acquisition.

Another important aspect of Fe homeostasis in *B. subtilis* is the formation of a biofilm. We previously showed that both the biofilm matrix and siderophore production are necessary for the organism to acquire Fe from precipitated oxides[22] and that biofilm-bound Fe can be used as a local Fe source[23]. Another study demonstrated the importance of Fe(III) in the deep layer of the biofilm, serving as a terminal electron acceptor in the absence of oxygen[24]. Biofilm-bound Fe thus appears essential for biofilm function in addition to cell growth. Siderophores and biofilm formation contribute to Fe acquisition and homeostasis in biofilm-forming bacteria[22,25]. However, little is known on the ecological advantage of secreting precipitating ligands such as PA in the context of microbial competition and the Fe's fate once complexed to PA.

In this work, we used a two-species bacterial model to investigate the role of pulcherrimin, if any, in the iron war. *Bacillus* and *Pseudomonas* are two genera often found together in the rhizosphere, where they interact with plant roots but also likely compete for the same resources. Indeed, many antagonistic interactions were demonstrated between *B. subtilis* and various *Pseudomonas* species[26,27]. However, less is known regarding nutrient acquisition, particularly Fe, during the competition between *Bacillus* and *Pseudomonas* genera. Here, using a two-species bacterial model, *Bacillus subtilis* (PA producer) and various *Pseudomonas* species, we show that *B. subtilis* produces PA in pairwise interaction with *Pseudomonas protegens* Pf-5, creating a local source of precipitated Fe(III) and is able to take up Fe from the pulcherrimin complex via the production of the strong Fe-chelator bacillibactin (BB). We also show that pulcherrimin mitigates oxidative stress in this organism, increasing survival and biofilm formation while in competition with Pf-5. Our observations provide a new understanding of the physiological role of pulcherrimin, which is neither a siderophore nor an inaccessible Fe sequester, but instead provides a local Fe source while preventing deleterious reactive oxygen species formation.

## Results

### *P. protegens* Pf-5 triggers pulcherriminic acid production in *B. subtilis*

To explore the defensive strategies used by *B. subtilis* in interaction with other soil bacteria, we performed time-course pairwise interaction assays between *B. subtilis* NCIB 3610 and three *Pseudomonas* species (*P. fluorescens* WCS365, *P. capeferrum* WCS358, and *P. protegens* Pf-5) isolated from the rhizosphere and used as biocontrol agents in crops[28,29]. These three *Pseudomonas* species were chosen for their differences in secondary metabolites production[30–32]. The interactions were examined on two different media: MSgg, commonly used to study biofilm formation in *B. subtilis* and which was previously used in *Bacillus* spp. – *Pseudomonas* pairwise interactions[27], and Murashige and Skoog medium (MS), commonly used to grow plants and examine their interactions with beneficial bacteria. Here, MS was supplemented with glycerol and glutamate to promote *B. subtilis* biofilm formation[33]. In MSgg Fe is provided as salt (FeCl$_3$; inorganic), which under oxic conditions forms ferric hydroxides and oxides that precipitate. In MS, Fe is provided as an organic complex (Fe-EDTA) which retains its

solubility through time. Nonetheless, in both conditions, Fe is not readily bioavailable for *B. subtilis*. Over time on MSgg, *B. subtilis* engulfed both *P. fluorescens* WCS365 and *P. capeferrum* WCS358, but *B. subtilis* and Pf-5 colonies did not establish a physical contact suggesting antagonistic activities mediated by diffusible molecules (Fig. 1a). Side-by-side colonies on MS display a lack of interaction, except for a red halo which surrounded *B. subtilis* colony in competition with Pf-5 (Fig. 1b). This red halo was absent when *B. subtilis* was alone on MS, but always present on MSgg, as described earlier[14] (Fig. 1a). Pulcherriminic acid (PA) is the sole known pigment produced by *B. subtilis*[14]. It is colorless but forms the red insoluble pulcherrimin complex upon Fe(III) binding outside the cell[34]. *B. subtilis* deleted for *yvmC*, a gene encoding the protein responsible for PA synthesis, challenged with Pf-5 on both media (MS and MSgg) lacked production of the red pigment under all conditions (Fig. 1a, b). Complementation of the *yvmC-cypX* operon at the *amyE* locus restored pigment synthesis (Supplementary Fig. 1a), demonstrating that the "red" phenotype is due to PA production.

On MSgg, *B. subtilis* displayed a strong and uniform expression of the P$_{yvmC}$-*yfp* bioreporter across the biofilm (Supplementary Fig. 2a, b). In contrast on MS, only *B. subtilis* cells next to Pf-5 displayed P$_{yvmC}$-*yfp* expression, and thus, PA production (Supplementary Fig. 2c). Flow cytometry analysis of the proportion of cells expressing P$_{yvmC}$-*yfp* confirmed a clear population shift when *B. subtilis* was grown close to Pf-5 with almost ~95% of YFP + , compare to only ~ 30% for *B. subtilis* colonies grown in the presence of WCS365 or WCS358 (Fig. 1c, d).

Pf-5 is known to secrete secondary metabolites detrimental to *B. subtilis*, including 2,4-DAPG and pyoluteorin[27,35], which could be responsible for triggering PA production. Challenging *B. subtilis* with Pf-5 Δ*phlD* mutant, unable to synthesize these two secondary metabolites[36], did not significantly decrease the number of cells expressing *yvmC* (P$_{yvmC}$-*yfp* reporter) (Supplementary Fig. 1c). However, cells were expressing P$_{yvmC}$-*yfp* to a lesser degree compared to WT when in presence of Pf-5, which suggests that either DAPG or pyoluteorin might be involved in PA production (Supplementary Fig. 1b). Spiking DAPG close to *B. subtilis* colony triggered P$_{yvmC}$-*yfp* expression (Supplementary Fig. 2d). Interestingly, after removing a slice of agar between 3610 and Pf-5 to eliminate the influence of medium-diffusible molecules, ~70% of cells in the population were still expressing P$_{yvmC}$-*yfp* (Supplementary Fig. 1c). Failing to directly pinpoint all the molecules responsible for triggering PA production, our observations suggest that it results from an effect of both volatiles and medium-diffusible secondary metabolites from Pf-5.

### Pulcherrimin is important for the survival of *B. subtilis* in interspecies interaction

On both media, in the absence of Pf-5, the WT and Δ*yvmC* (PA mutant) displayed no significant differences in biofilm phenotypes, growth curve, and biomass (Fig. 2a, b) except on MSgg at later time points where Δ*yvmC* kept expanding as previously observed (Fig. 1a, b)[14]. However, Pf-5 impeded biofilm formation in WT on both media, and this effect was exacerbated in Δ*yvmC* mutant, observed by a lack of wrinkles in the presence of Pf-5 (Fig. 1a, b). Furthermore, Pf-5 completely stopped the growth and survival of Δ*yvmC* (Fig. 2c, d). Together, these findings demonstrated that secretion of PA protects *B. subtilis* cells and allows biofilm formation against Pf-5.

### Pf-5 and pulcherrimin trigger P$_{dhbA}$-*lacZ* expression, and Pf-5 presence increases Fe load in *B. subtilis* biofilm

Secretion of a strong and insoluble Fe chelator such as pulcherriminic acid could affect Fe accumulation in the biofilm. Thus, the total Fe content present in the colonies (biofilms + cells) and cells (intracellular Fe content) of *B. subtilis* WT and Δ*yvmC* mutant (PA mutant) was quantified. There was a slight diminution of Fe accumulation in Δ*yvmC* biofilm (1.58E−7 M) compared to the WT alone (2.92E−7 M) in MSgg

medium when PA is highly produced (Fig. 3a), while the intracellular Fe contents remained similar (Supplementary Fig. S3a). On the same medium, *B. subtilis* WT in the presence of Pf-5 showed ∼ 3-fold increase in Fe content in the colonies (biofilms + cells) compared to WT alone (8.45E−7 M), but the intracellular Fe contents did not differ significantly (Supplementary Fig. S3a). Also, the Δ*yvmC* mutant in presence of Pf-5 accumulated significantly more Fe in the colonies (3.20E−7 M) compared to Δ*yvmC* (1.58E−7 M), but significantly less than the WT next to Pf-5 (Fig. 3a). These results imply that pulcherrimin production favors the extracellular accumulation of Fe in the biofilm. This observation was confirmed by an increased Fe accumulation in MS in the WT biofilm in the presence of Pf-5, which stimulates PA production in the MS medium (Fig. 3b). In MS only, the difference between PA producer and non-producer was not significant in both the colonies and intracellularly (Fig. 3b; Supplementary Fig. S3b).

The maintenance of intracellular Fe homeostasis (Supplementary Fig. S3) despite the formation of the insoluble PA-Fe complex suggests the presence of a strong siderophore production. To examine this hypothesis, a *B. subtilis* strain harboring P$_{dhbA}$-*lacZ*, a β-galactosidase transcriptional reporter of the DHB and BB biosynthesis operon, was used. A colony grown for 24 h on MSgg displayed a small blue ring in its

center, and this ring rapidly darkened over time, suggesting Fe limitation as a result of Fe precipitation (Fig. 3c). Interestingly, after 96 h P$_{dhbA}$-*lacZ* signal was also visible at the outer ring of the WT but not in Δ*yvmC* (Fig. 3c), which likely results from Fe sequestration by pulcherrimin as described in an earlier study[14]. The P$_{dhbA}$-*lacZ* expression was also quantified at 48 h on MSgg where the WT showed increased level of LacZ activity in presence of Pf-5 when PA is produced abundantly (Fig. 3e). The Δ*yvmC* mutant showed reduced LacZ activity compared to WT, but an increase was observed next to Pf-5 suggesting siderophore production by the competing colony (Fig. 3e). In MS medium, where Fe is kept in solution by EDTA, P$_{dhbA}$-*lacZ* appeared very weakly expressed (Fig. 3d). In the presence of Pf-5, on both media, the WT displayed a dark blue coloration (Fig. 3c, d). A clear physical gradient of Fe starvation could be observed on WT with Pf-5 in MS medium since the cells facing the Pf-5 colonies were a much darker hue than the rest of the colony (Supplementary Fig. 4a). The P$_{dhbA}$-*lacZ* expression was also quantified at 24 h on MS medium to confirm that the P$_{dhbA}$-*lacZ* expression was increased only when PA is highly produced (Fig. 3f). To validate the contribution of pulcherrimin in stimulating *dhbA* transcription, the mutant Δ*pchR* (the negative regulator of PA synthesis) was used. As expected, Δ*pchR* experienced severe Fe

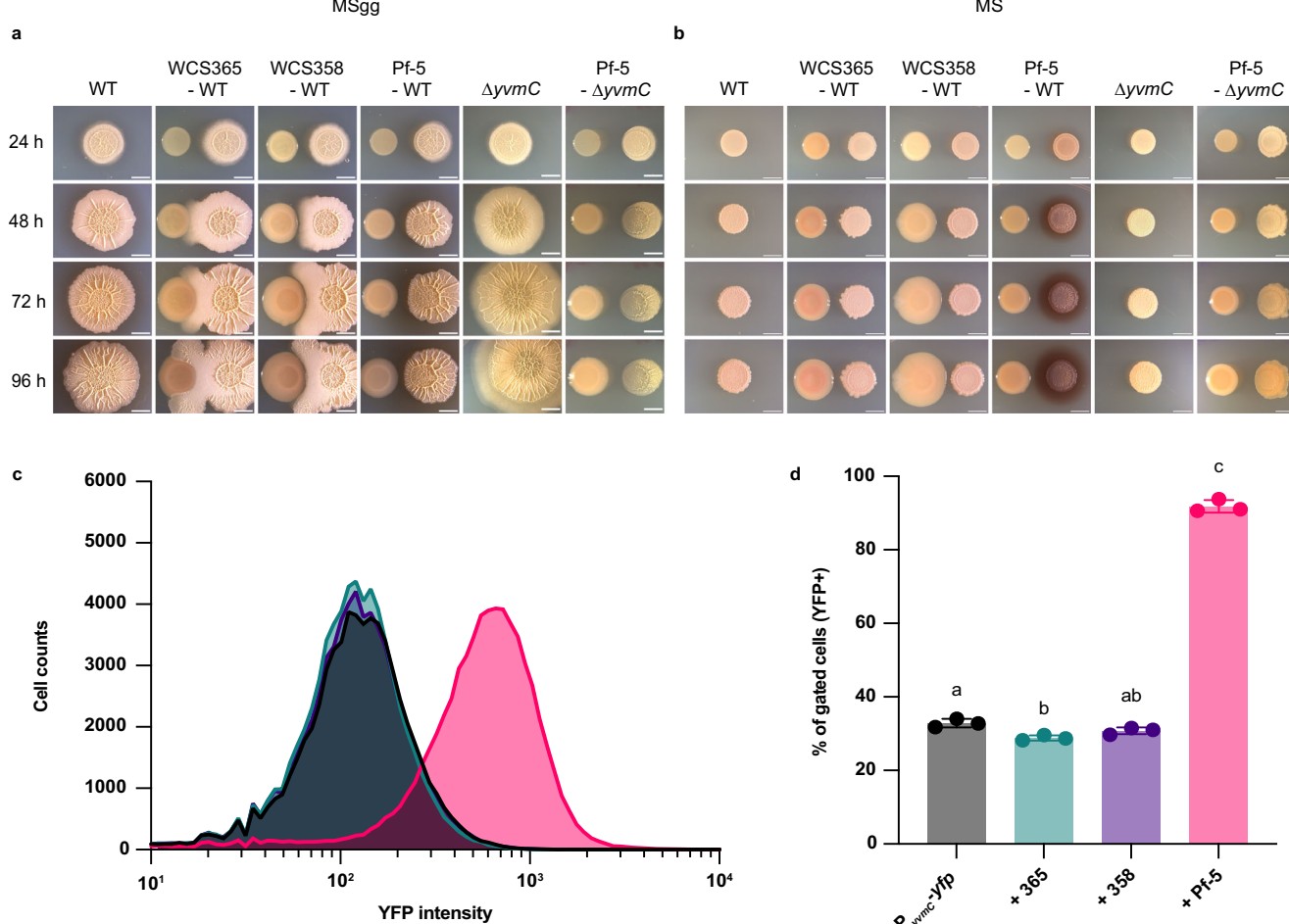

**Fig. 1 | P. protegens Pf-5 triggers pulchirriminic acid secretion in B. subtilis.**
**a** Representative images of biofilms from *B. subtilis* WT and Δ*yvmC*, in competition with *Pseudomonas* WCS365, WCS358, and Pf-5. Pictures were taken at regular intervals on MSgg and on (**b**), MS medium. Images contrast was adjusted to allow clear visualization. **c** Flow cytometry analysis showing the distribution of fluorescence intensity of YFP-based transcriptional reporter for *yvmC* alone (black) against *Pseudomonas* species WCS365 (teal), WCS358 (purple), and Pf-5 (pink). **d** *B. subtilis* harboring the fluorescent reporter for PA production (P$_{yvmC}$ - *yfp*) was grown on MS

for 24 h with and without *Pseudomonas* species. The percentage of fluorescent cells (YFP + ) was analyzed by flow cytometry. All experiments were performed in three biological replicates with three technical replicates. Representative experiments and pictures are presented. Different letters indicate statistically significant differences, $P < 0.05$, one-way ANOVA and Tukey's multiple comparisons test. Data are presented as mean values ± SD, $n = 3$. Scale bar, 5 mm. Source data are provided as a Source Data file.

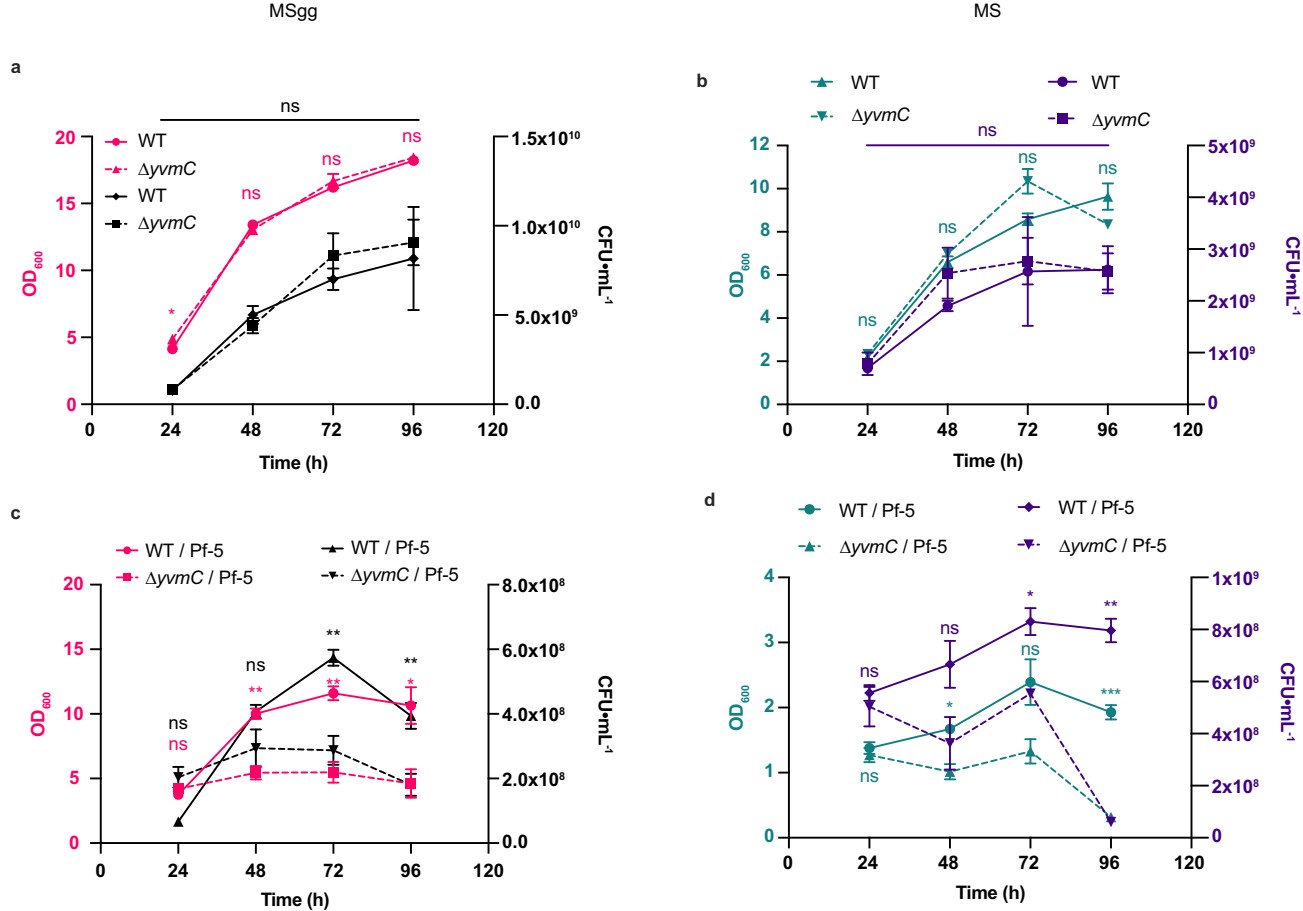

**Fig. 2 | Pulcherrimin is important for *B. subtilis* survival and biofilm formation.**
**a** CFU counts and biomass (OD$_{600}$) of WT and Δ*yvmC* alone on MSgg (ns non-significant, *$P = 1.82 \times 10^{-2}$) and (**b**), on MS (ns non-significant). **c** CFU counts and biomass (OD$_{600}$) of WT and Δ*yvmC* with Pf-5 on MSgg (ns non-significant, ** = $P < 0.01$) and (**d**), on MS. (ns non-significant, *$P < 0.05$, **$P < 0.01$, ***$P = 8.93 \times 10^{-4}$, two-way ANOVA, sidak's). All experiments were performed in three biological replicates with three technical replicates. Representative experiments are presented. Data are presented as mean values ± SD, $n = 3$. Source data are provided as a Source Data file.

starvation, as demonstrated by a strong expression of *lacZ* under and on top of the biofilm (Supplementary Fig. 4a–c). Moreover, even if pulcherrimin accumulation triggered *dhbA* expression, the presence of Pf-5 was necessary to achieve strong *dhbA* expression throughout the biofilm (Supplementary Fig. 4b). These findings demonstrate that a combination of Pf-5 and pulcherrimin production creates a Fe stress eliciting strong P$_{dhbA}$-*lacZ* expression by *B. subtilis* cells suggesting more DHB and BB synthesis.

**Bacillibactin siderophore is key to mobilizing pulcherrimin-bound Fe**
The maintenance of WT growth in the presence of Pf-5 in MS, despite the fact that Fe(III) is chelated in the insoluble pulcherrimin complex, raises questions on how *B. subtilis* acquires Fe under those conditions. Hence, we present a conceptual framework summarizing two possible ways for Fe recovery in the presence of pulcherrimin (Fig. 4a). First, the PA-Fe complex could be taken up by *B. subtilis* cells, confirming its role as a siderophore as previously suggested[19]. Alternatively, Fe might be remobilized by the strong BB siderophore, which is overproduced under these conditions. Similarly, Pf-5 might retrieve Fe from pulcherrimin owing the production of the proper tool (i.e., the high-affinity Fe ligand pyoverdine). As seen in Fig. 4b, the addition of either FeCl$_3$ or pulcherrimin restored *B. subtilis* WT growth (monitored as OD$_{600}$) compared to Fe depleted condition. Repeating this experiment with a Δ*dhbF* mutant, unable to catalyze the last step of BB synthesis, demonstrated a clear growth defect in the presence of pulcherrimin as

the sole Fe source but not in the presence of FeCl$_3$ in accordance with a previous study (Fig. 4c)[12,24]. Finally, a Δ*dhbA-F* mutant, which is unable to produce neither DHB nor BB, could not grow in presence of pulcherrimin and FeCl$_3$ but was able to growth in presence of Fe-citrate (Supplementary Fig. S5). This result suggests that pulcherrimin is not a siderophore per se and that siderophores are required for the acquisition of pulcherrimin-bound Fe. Similarly, *B. subtilis* WT, but not the Δ*dhbF* mutant, can use pulcherrimin as a sole Fe source for forming the Fe-rich pellicle (Fig. 4e), a process that requires a large amount of Fe[37].

Since sequestering Fe in pulcherrimin does not prevent Fe acquisition by *B. subtilis*, it might starve Pf-5 by monopolizing Fe (Fig. 4a). Surprisingly, Pf-5 still displayed strong growth in the presence of either pulcherrimin or FeCl$_3$ compared to MSgg without Fe (Fig. 4d), showing that pulcherrimin-bound Fe can be used by the competing Pf-5.

To confirm that pulcherrimin-bound Fe(III) was remobilized by BB, in vitro dissociation assay was performed. This experiment requires high-quality purified compounds, which is not attainable with the harsh steps of the pulcherrimin purification procedure. Hence, the apo-PA was synthesized to obtain the soluble molecule (see supplementary data for synthesis details; see supplementary Fig. S6 for chemical structures). The addition of Fe(III) to the apo-PA led to the formation of a red, flaky precipitate at a 3:1 ratio (PA: Fe) (Fig. 5c). First, pulcherrimin (i.e., Fe-PA) was mixed with apo-BB in a 1:1 ratio based on Fe binding capacity, and the evolution of Fe binding by BB and re-solubilization of PA due to dissociation over

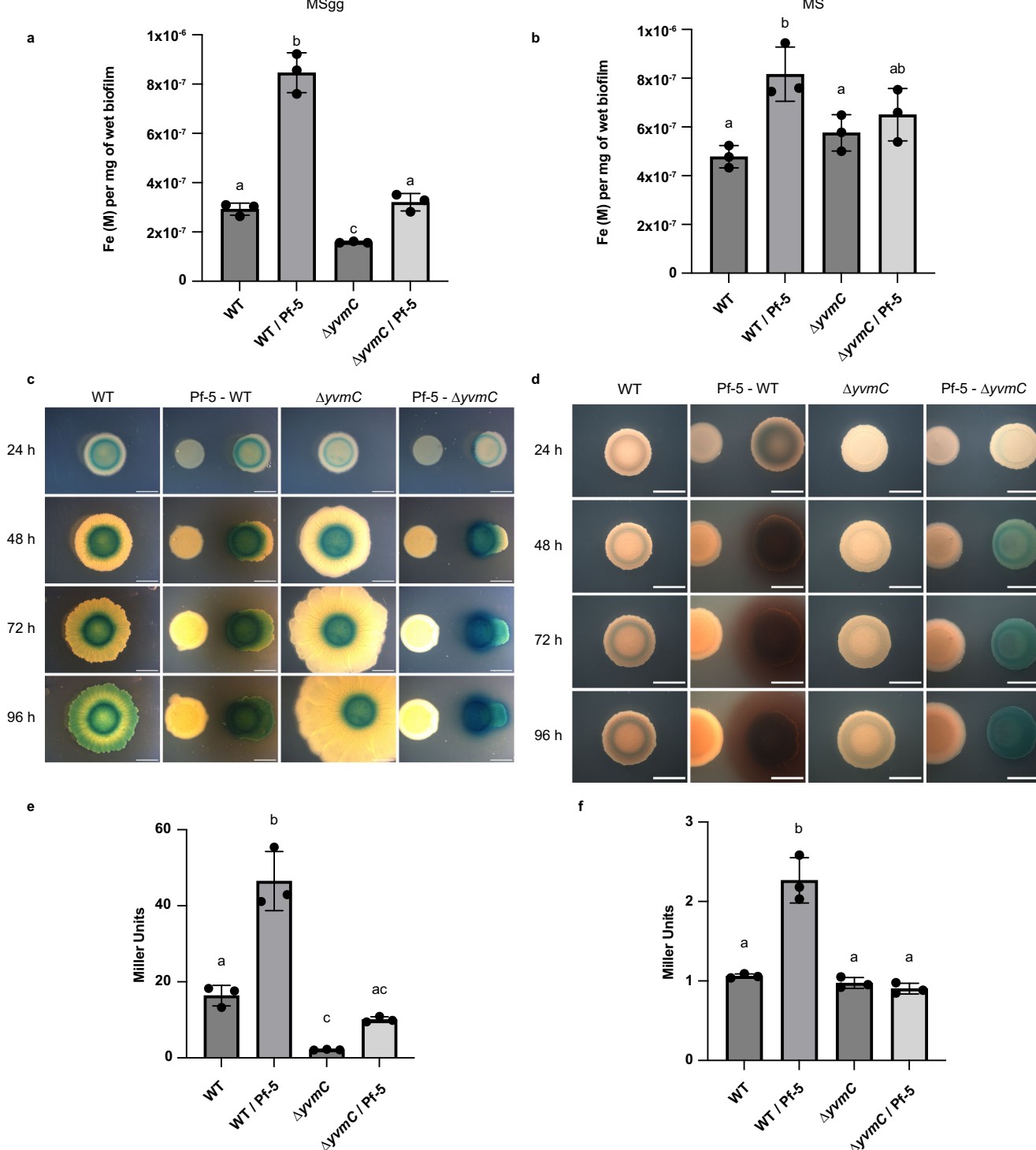

**Fig. 3 | Pf-5 and pulcherrimin trigger bacillibactin synthesis and Pf-5 presence increases Fe load in *B. subtilis* biofilm. a** Fe content (in molar; M), normalized on mg of wet biofilms, of 48 h colonies formed on MSgg and (**b**), on MS medium. **c** Bottom-view pictures of *B. subtilis* harboring the β-galactosidase gene transcriptional reporter for DhbA production ($P_{dhbA}$-*lacZ*) grown on MSgg and (**d**), on MS medium. Images contrast was adjusted to allow clear visualization. **e** β-galactosidase activities of WT and Δ*yvmC* harboring the $P_{dhbA}$-*lacZ* reporter alone and in interaction with Pf-5 in MSgg at 48 h (**f**), and in MS at 24 h. Experiments were performed in three biological replicates with three technical replicates. Different letters indicate statistically significant differences, $P < 0.05$, one-way ANOVA and Tukey's multiple comparisons test. Data are presented as mean values ± SD, $n = 3$. Scale bar, 5 mm. Source data are provided as a Source Data file.

time was followed using UPLC-MS. As seen in Fig. 5a, a decrease of apo-BB and an increase of apo-PA were observed. The apo-PA in absence of BB was quantified every 24 h until 96 h and the pulcherrimin precipitates were weighted at T = 0 and T + 96 h to evaluate their stability. No significant changes were observed

(Supplementary Fig. S7), confirming that BB dissociated pulcherrimin by binding Fe and solubilizing the precursor, PA. The size of pulcherrimin aggregates also decreased over time and can be visualized (Fig. 5c). In parallel, the dry precipitate was weighted, and a decrease in pulcherrimin mass was observed, going from

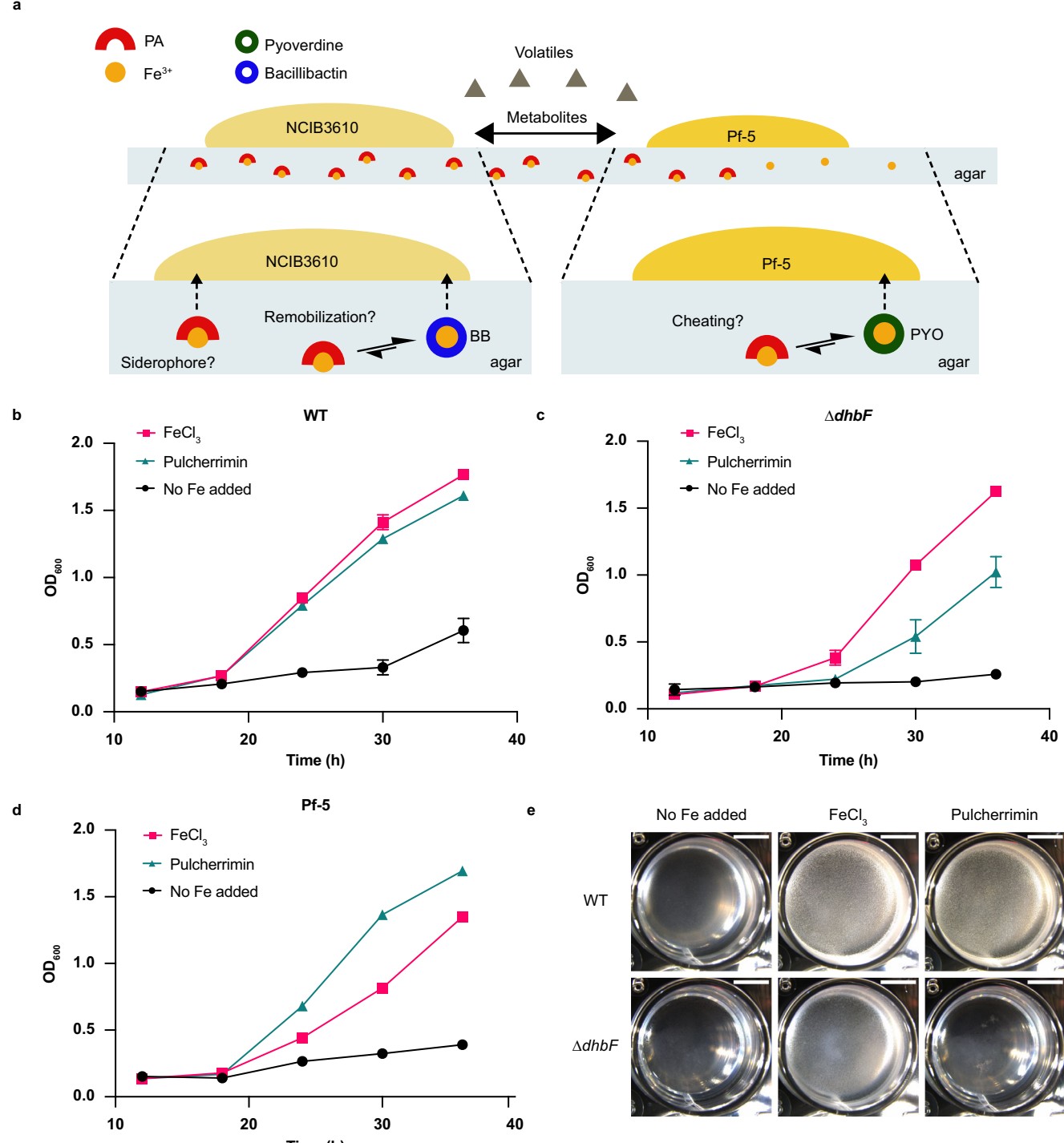

**Fig. 4 | Bacillibactin siderophore is key to mobilizing pulcherrimin-bound Fe.**
**a** Conceptual framework in MSgg showing the possible route of Fe recovery. Pulcherrimin could be uptake as a complex (siderophore), or Fe trapped in pulcherrimin could be remobilize by a stronger siderophore (BB) based of affinity constant. A foreign species (Pf-5) could strip Fe from pulcherrimin and used it as an Fe source. **b,** *B. subtilis* WT growth was monitored (by absorbance at $OD_{600}$) in MSgg medium. Optical density was measured every 6 h starting at 12 h post-inoculation (12 h, 18 h, 24 h, 30 h, and 36 h) in presence of no added Fe (control), 50 μM $FeCl_3$, and 50 μM pulcherrimin as Fe sources. **c** The growth of Δ*dhbF* was

monitored as in (**a**). **d** The competitive strain *P. protegens* Pf-5 growth was evaluated as in (**a**). **e** Pellicle formation assays were performed to evaluate the biofilm robustness in the presence of no added Fe, 50 μM $FeCl_3$, and 50 μM pulcherrimin as Fe sources. Pellicles were cultivated in a 24-well plate and pictures were taken 24 h later. Scale bar (top upper right−5 mm). All experiments were performed in three biological replicates with three technical replicates. Representative experiments and pictures are presented. Data are presented as mean values ±SD, *n* = 3. Source data are provided as a Source Data file.

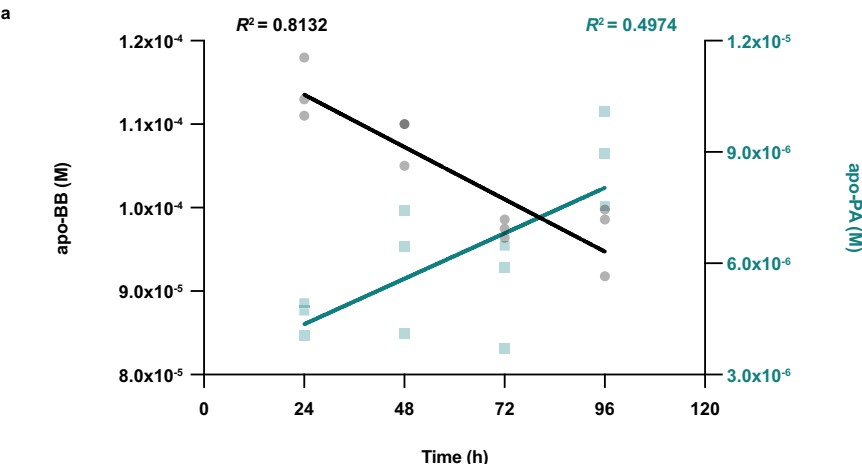

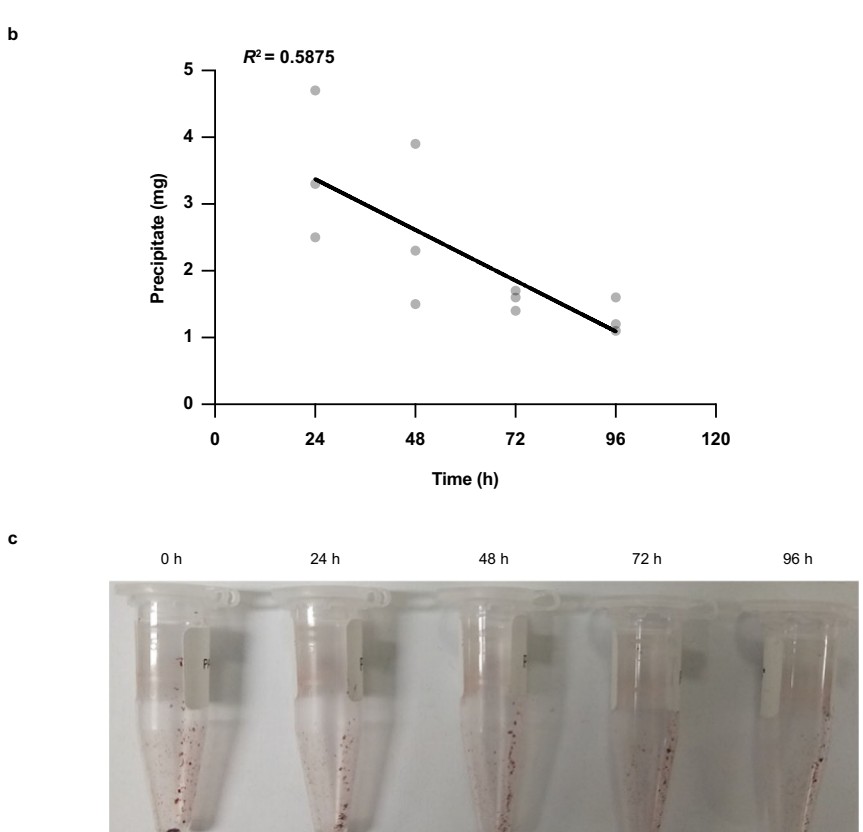

**Fig. 5 | Bacillibactin mobilizes pulcherrimin-bound Fe. a** Apo-BB was mixed with precipitated pulcherrimin at a 1:1 ratio. Apo-siderophores (BB and PA) were quantified by UPLC-MS and samples were collected at 24 h, 48 h, 72 h, and 96 h. $n = 12$. **b** Tubes were centrifuged and dried to determine the mass of the precipitate over time. $n = 12$. **c** Images were taken at 0, 24 h, 48 h, 72 h, and 96 h to visualize pulcherrimin dissociation. $n = 3$. Source data are provided as a Source Data file.

$3.50 \pm 1.11$ mg to $1.30 \pm 0.26$ mg after 96 h confirming the dissociation of pulcherrimin by BB (Fig. 5b).

## Immobilization of Fe(III) by PA prevents oxidative stress in the biofilm

*B. subtilis* requires a large amount of Fe for biofilm formation, but Fe overload is known to cause detrimental effects to cells in the presence of oxygen[38]. Consequently, tight regulation is needed to prevent the reduction-oxidation (redox) cycle and the Fenton and Fenton-like reaction occurring in these conditions. The precipitation of pulcherrimin creates a large pool of local Fe(III), accessible through BB complexation (see above), while possibly preventing the accumulation of

reactive oxygen species (ROS). ROS levels were quantified extracellularly and intracellularly in the WT and the $\Delta ywmC$ mutant using the 2',7'-dichlorodihydrofluorescin diacetate (DCFH$_2$-DA) probe. DCFH$_2$-DA is a widely used fluorescent probe for the detection of general oxidative stress[39,40]. As seen in Fig. 6a, b, WT biofilms showed low ROS levels on MSgg, where PA is secreted in high amounts, and on MS medium, where PA production is minimal. In contrast, the $\Delta ywmC$ mutant showed high ROS levels in MSgg but not in MS medium, indicating that PA production and subsequent pulcherrimin formation are responsible for controlling oxidative stress on MSgg both extracellularly and intracellularly (Supplementary Fig. S8a–d). Complementation in *trans* of the $\Delta ywmC$ reduced ROS levels similar to WT (Fig. 6a, b).

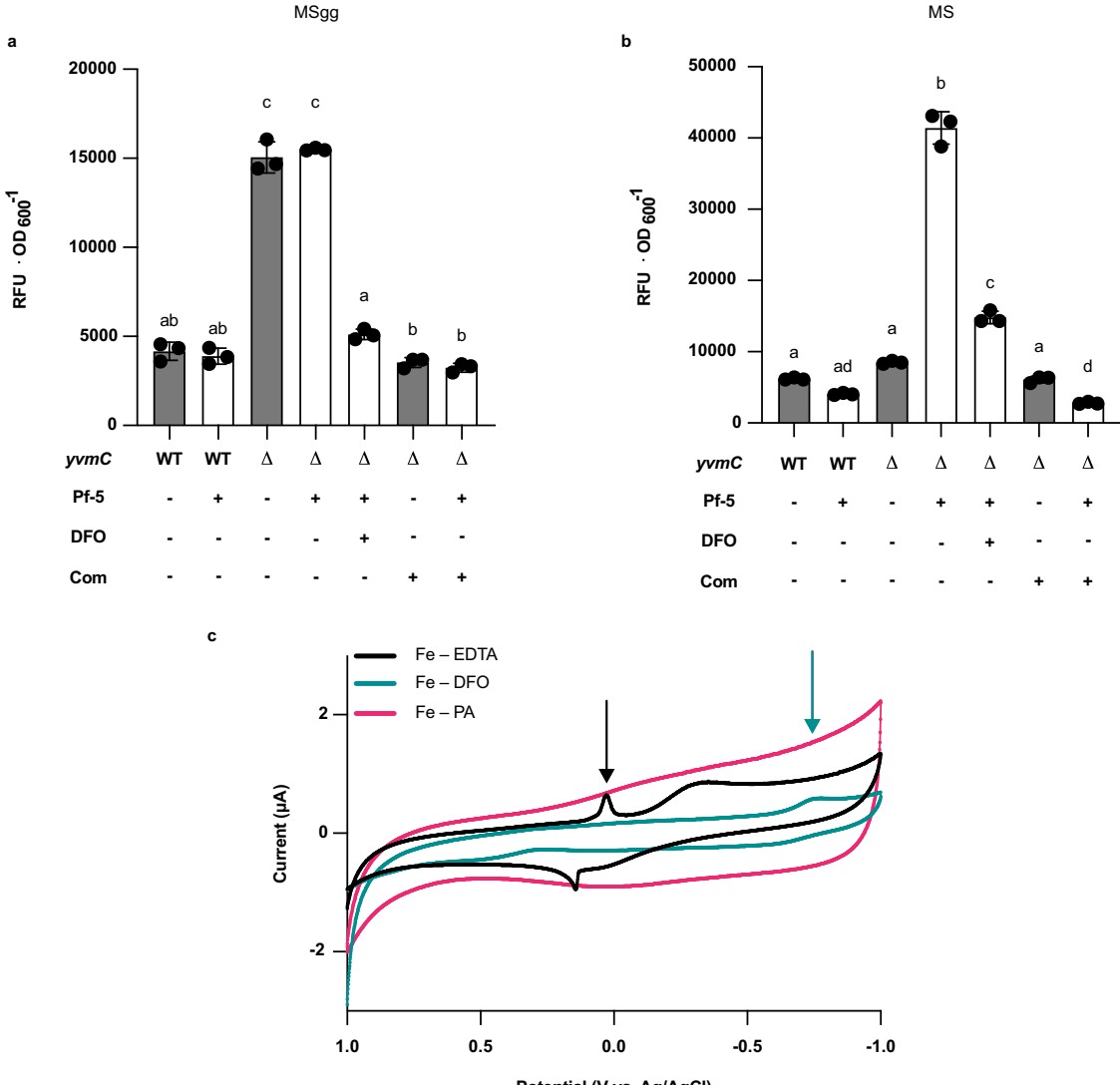

**Fig. 6 | Pulcherrimin prevents oxidative stress in biofilm by immobilizing Fe(III). a** Quantification of general oxidative stress (ROS) using the DCFH$_2$-DA probe in MSgg medium. Fluorescence intensities were normalized on biomass (OD$_{600}$). DFO indicates that the siderophore deferoxamine mesylate was added to chelate Fe; Com indicates the complementation by $amyE$::P$_{yvmC}$-$yvmC$-$cypX$ in the $\Delta yvmC$ mutant background. Experiments was performed in three biological replicates with three technical replicates. **b** Same as in (**a**), except that the experiment was conducted in MS medium. Experiments was performed in three biological replicates with three technical replicates. Data are presented as mean values ±SD, $n = 3$. Different letters indicate statistically significant differences, $P < 0.05$, one-way ANOVA and Tukey's multiple comparisons test. Gray strips indicate absence of Pf-5 and white strips indicate presence of Pf-5. **c** Cyclic voltammetry profile curves of Fe(III)-EDTA (black line; the arrow indicates the Fe reduction peak), Fe(III)-DFO (teal line; the arrow indicates the Fe reduction peak, DFO denotes Deferoxamine) and pulcherrimin (PA-Fe(III); pink line). Source data are provided as a Source Data file.

When grown on MS medium, PA is produced by the WT at a high level only when Pf-5 is present. Also, $\Delta yvmC$ but not WT cells showed high ROS levels in the presence of Pf-5. These results suggest that the presence of Pf-5 creates an oxidative environment that is mitigated by pulcherrimin. To confirm that PA prevents oxidative stress through Fe-sequestration, the resuspended biofilms were chemically supplemented with deferoxamine (DFO), a well-known ROS protective agent[41] and TPEN (N,N,N',N'-Tetrakis(2-pyridylmethyl)ethylenediamine), a metal chelators with high affinity for zinc[42,43]. Both DFO and TPEN supplementation in the $\Delta yvmC$ mutant reduce extracellular ROS levels comparable to WT in both media (Fig. 6a, b). However, TPEN did not reduce intracellular ROS at levels comparable to DFO supplementation (Supplementary Fig. S8c, d) which implies that pulcherrimin, through Fe precipitation, mitigated oxidative stress in *B. subtilis*.

One distinctive trait of PA compared to most Fe-chelating agents is that after Fe binding, the resulting pulcherrimin complex precipitates. To explore the idea that Fe-sequestration by pulcherrimin immobilizes Fe(III) and prevents Fenton reactions, the redox potential of Fe bound to PA was compared with two known metals chelators: EDTA and deferoxamine. The pre-formed complex Fe-EDTA showed reversible oxidation and reduction peaks centered at E = 83 mV vs. Ag/AgCl (Fig. 6c). The Fe-complexation by the strong chelator deferoxamine (pre-formed), in contrast, induced a large cathodic shift in the reduction potential to −750 mV. This result demonstrates that reducing Fe bound to deferoxamine is more difficult (Fig. 6c). For pulcherrimin compared to all ligands, no electrochemical reaction has been measured, which could either suggest: (1) that it binds to Fe(III) and precipitates which leads to a concentration below our detectable voltammetric levels; or (2) that Fe(III)-bound pulcherrimin possesses a reduction potential outside the window we scanned in (Fig. 6c; Supplementary Fig. S9). Nevertheless, both hypotheses lead to low Fe reduction in comparison to the other Fe(III)-chelators, demonstrating that pulcherrimin strongly counteract oxidative stress.

## Discussion

In natural environments, Fe acquisition is of central importance for microbes to compete and protect their ecological niches. Knowing the various strategies and molecules involved in Fe scavenging and sequestration is necessary to understand what drives bacterial interactions in natural communities. This study provides an unprecedented understanding of how pulcherrimin, a Fe-precipitating molecule from *B. subtilis* and other microorganisms, is involved in Fe acquisition and acts as a protective agent.

While PA production by *B. subtilis* and *Metschnikowia* was suggested to be either constitutive or modulated by abiotic factors[14,44], we report here for the first time that it can be secreted in response to a specific competitor making PA part of *B. subtilis* competition toolbox. Signals triggering PA secretion are likely multifactorial since both diffusible molecules and volatiles produced by Pf-5 trigger a high proportion of PA-producing cells in the *B. subtilis* population (Fig. 1). There is multiple evidence that volatiles play an important role in modulating microbial communities[45,46]. Notably, *Pseudomonas* species' volatiles are diverse, and they can antagonize other microorganisms, including fungal and bacterial plant pathogens[47–49]. It could be interesting to examine how the production of pulcherrimin affects the fitness of *B. subtilis* in presence of other bacterial species and more importantly if pulcherrimin is produced during plant roots colonization.

The most striking trait of pulcherriminic acid is its precipitation upon Fe(III) binding[15]. While pulcherrimin has been proposed to act as a siderophore[19], our work indicates that the PA-Fe complex is not readily available for uptake and is thus not a siderophore per se. However, our data also suggest that it could be involved in a larger strategy for Fe acquisition. One major limitation of siderophore-assisted acquisition is the high cost of siderophore production and Fe-siderophore complex retrieval. Furthermore, the loss of siderophores before recovery of Fe due to diffusion away from the cells and degradation (e.g., some siderophores are sensitive to oxidation) is likely to add to the cost of Fe uptake. PA synthesis is a two-step enzymatic reaction using Leucine-tRNA as precursor[50], and BB is produced via a four-step non-ribosomal peptide synthesis[12]. Thus, the metabolic cost of PA is significantly lower than the one of BB. We propose that pulcherrimin could be part of a "store and mine" strategy. We can envision a model in which PA (low-metabolic cost) is secreted to rapidly immobilize environmental Fe within and in the proximity of the biofilm. This immobilized local source of Fe, unaffected by potential flows, could then be retrieved by BB (Fig. 4) when Fe is needed.

The sequestration of valuable nutrients, intracellularly and extracellularly, is an effective strategy to manage competition[51,52]. Sequestering Fe(III) in a precipitate would be very effective in slowing Fe access to neighboring competitors lacking strong siderophores, providing a significant competitive edge to *B. subtilis*. But, as observed with Pf-5 (Fig. 4d), it will not inhibit Fe acquisition by competitors owning high-affinity siderophores. In soils where multiple stressors (e.g., nutrients) constrain metabolic activity, immobilizing Fe still bear merits, if it results in an overall higher tax on Fe acquisition by the competitor than *B. subtilis*. Thus, we argue that, while extracellular Fe sequestration by PA is most likely not a one fits all strategy to manage competitors, it is another valuable weapon in the arsenal at the disposal of *B. subtilis*, and potentially other species, to manage Fe when under competitive stress. Further research on PA-Fe use efficiency by *B. subtilis* and competitors is needed to further test this hypothesis.

It is well established that in addition to Fe acquisition, siderophores have multiple physiological roles which result from their Fe chelation abilities[53,54]. In addition to its role as a Fe immobilizer, we observed that pulcherrimin could reduce oxidative stress levels in *B. subtilis* biofilm. Oxidative stress and Fe are interconnected via the Fenton reaction, in which Fe(II) is the substrate and Fe(III) is the product. Fe(III) binding by chelating agents, such as siderophores

produced by various bacterial species, can impede its reduction to Fe(II) and thus prevent Fenton reaction[53,55,56]. By doing so, pulcherrimin would limit mortality rates due to ROS and allow sustained biofilm formation (Fig. 2) since it was shown that *B. subtilis* needs to tightly control the level of ROS in order to fully develop its biofilm[57]. Using cyclic voltammetry, we confirmed that immobilization of Fe(III) by pulcherrimin prevents its reduction. Notably, the weak siderophore DHB, produced in high concentrations by *B. subtilis* against Pf-5, is a phenolic compound that can promote the Fenton reaction[58,59]. Hence, we propose that PA production and subsequent pulcherrimin complex formation counteract the effect of DHB (Fig. 3c, d) and protect *B. subtilis* against oxidative stress during biofilm formation since it requires a large production of DHB[22,24] as well as accumulation of Fe for metabolic activities in the anoxic layers of the biofilm[24].

Pulcherrimin and siderophore production are traits shared by many microbial species, and the Fe management strategies uncovered in this study may apply to more microbes other than *B. subtilis*. Indeed, PA has been confirmed, through comparative genome analysis, to be produced by isolates of *Bacillus cereus* (opportunistic pathogen) and *Staphylococcus epidermidis* (skin commensal bacterium)[14]. Forming an immobilized but accessible local source of Fe controlled by Fe-precipitating molecules and siderophore production is an appealing strategy for biofilm (or colony) scale management of Fe and could be widespread in many ecosystems.

## Methods

### Strains and culture conditions

The strains and plasmids used in this study are listed in Table S1. *B. subtilis* strains were routinely cultivated in lysogenic broth (Luria-Bertani LB; 1% w/v tryptone, 0.5% w/v yeasts extract, 0.5% w/v NaCl) at 37 °C in agitation for 3 h. *Pseudomonas* strains were cultivated in LB at 30 °C in agitation for 3 h. When necessary, antibiotics were used at the following concentration: spectinomycin (100 μg·mL$^{-1}$), kanamycin (10 μg·mL$^{-1}$), ampicillin (100 μg·mL$^{-1}$).

### Strains construction

All deletion mutants used in this study were purchased from the Bacillus Genetic Stock Center (BGSC) collection (http://www.bgsc.org) in the *B. subtilis* 168 background and were introduced in the undomesticated strains NCIB 3610 by transduction using SPP1-mediated generalized transduction[60]. For the P$_{ywmC}$-*yfp* transcriptional reporters, the promoter region of *ywmC* was amplified by PCR from NCIB 3610 gDNA using primers PBB666 (5′-AGTCGAATTCTGTTCATTAAGGTGCA GCAGTCTCAC-3′) and PBB667 (5′- GTAGCATGCCTATTATGCCCCGTC AAACGCAACG-3′). The PCR fragment was digested using the EcoRI and SphI restrictions enzymes and was ligated into the plasmid pKM003[61]. The resulting plasmid was linearized and transformed in *B. subtilis* 168 before being introduced into NCIB 3610 by transduction using SPP1 phages. For the P$_{dhbA}$-*lacZ* transcriptional fusion, the promoter region of *dhbA* was amplified by PCR from NCIB 3610 gDNA using primers PBB696 (5′- GCTAGAATTCGTATACGGGCAGAATTTTGCGAGT-3′) and PBB697 (5′- GTACGGATCCTGCGCCTTGACTGGCAAGC-3′). Primers were ordered from Integrated DNA Technologies, IDT. The PCR fragment was digested using the EcoRI and BamHI restrictions enzymes and ligated into the plasmid pDG1728[62]. The resulting plasmid was linearized and transformed in *B. subtilis* 168 before being introduced into NCIB 3610, Δ*ywmC*, and Δ*pchR* by transduction using SPP1 phages.

### Pairwise interactions

*B. subtilis* strains and *Pseudomonas* species were washed in PBS, and their OD$_{600}$ was adjusted at 0.6. Ten μL of cell suspension was then spotted at a 0.7 cm distance onto Murashige and Skoog (MS; Sigma – M5519; 20.61 mM NH$_4$NO$_3$, 100 μM H$_3$BO$_3$, 2.99 mM CaCl$_2$, 0.11 μM CoCl$_2$·6H$_2$O, 0.1 μM CuSO$_4$·5H$_2$O, 100 μM Na$_2$-EDTA, 100 μM FeSO$_4$·7H$_2$O, 1.5 mM MgSO$_4$, 100 μM MnSO$_4$·H$_2$O, 1.03 μM Na$_2$MoO$_4$·2H$_2$O, 5 μM KI, 18.79 mM

$KNO_3$, 1.25 mM $KH_2PO_4$, 29.91 μM $ZnSO_4 \cdot 7H_2O$, 26.64 μM glycine, 0.56 μM myo-inositol, 4.06 μM nicotinic acid, 2.43 μM pyridoxine·HCl, 0.30 μM thiamine·HCl, 0.5% v/v glycerol, and 0.5% v/v glutamate) or MSgg (5 mM potassium phosphate and 100 mM MOPS (3-(N-morpholino) propanesulfonic acid) at pH 7.0 with 2 mM $MgCl_2$, 700 μM $CaCl_2$, 50 μM $MnCl_2$, 50 μM $FeCl_3$, 1 μM $ZnCl_2$, 2 μM thiamine, 0.5% v/v glycerol, and 0.5% v/v glutamate) supplemented with 1.5 % w/v agar, and incubated at 30 °C. When needed, X-Gal (5-Bromo-4-Chloro-3-Indolyl β-D-Galactopyranoside) was added at a final concentration of 120 μg·mL⁻¹. To assess biomass and CFUs, B. subtilis biofilms were removed using a sterile P200 tip and transferred into an Eppendorf tube containing 1 mL PBS. Biofilms were sonicated for 30 s to 60 s (until homogeneity) without pause at 30% amplitude. Cells were diluted and plated on LB for CFUs counting. Optical density at 600 nm (Biomass) was evaluated with a Genesys 30 S UV-Vis spectrophotometer.

## Microscopy

Two microscopes were used for this study. The stereo microscope Leica M165 FC was used to take brightfield images at different time points using the Leica MC170 HD camera. For fluorescence images, the Leica DFC3000 G (grayscale) camera was used. The Zeiss microscope Axio Zoom. V16 equipped with AxioCam 506 color camera was also used. Fiji (ImageJ1; version 2.9.0/ 1.53t) was used for image analysis.

## Flow cytometry analysis

For flow cytometry analysis of the amyE::P_yvmC-yfp reporter, cells grown for 24 h were removed from the agar surface with a P200 tip and placed into an Eppendorf tube containing 1 mL of PBS. Cells were sonicated with an amplitude of 30% between 30 and 60 s without pause until homogeneity, then centrifuged at 13,800 × g for 2 min before adding 200 μL of paraformaldehyde 4%. Cells were fixed for 7 min and washed twice with PBS before analysis. Data collection was performed with BD FACSJazz using a 488-nm laser and BD Accuri C6 plus. Data analysis was achieved with BD software (BD CSampler plus version 1.0.23.1 and BD FACS Sortware 1.2.0.142).

## Iron quantification by ICP-MS

Biofilms were weighed and then digested using 0.8 mL of nitric acid (trace metals grade; Fischer Scientific) on an SCP science Digiprep Jr at 65 °C for 45 min. After the digestion, 0.4 mL was transferred into a 15 mL falcon tube and filled with Milli-Q water to reach a final volume of 10 mL. For intracellular Fe quantification, cells were isolated from the matrix from an optimized protocol develop elsewhere[22]. Total biofilms were recovered in 1 mL of oxalate/EDTA (0.1 M/0.05 M) and incubated at room temperature (RT) for 7 min. Samples were centrifuged at 6500 × g for 7 min. The supernatant was discarded, and cells were resuspended in 1 mL NaCl 0.5 M before being sonicated for 30 s at 30% amplitude. NaOH was added to the supernatant at a final concentration of 0.1 M and incubated for 5 min at RT. Samples were centrifuged at 6500 × g for 7 min and resuspended again in 1 mL of oxalate/EDTA (0.1 M/0.05 M) for 7 min. Samples were centrifuged like the previous steps and pellets were resuspended in diluted oxalate/EDTA solution (0.025 M/0.0125 M) for 7 min. Finally, cells were centrifugated and stored at 4 °C until analysis. Cells were digested as described above. Samples were analyzed for Fe content on an ICP-MS XSeries 2; Thermo Scientific and on an ICP-MS Agilent 7850 equipped with an autosampler SPS 4.

## Pulcherrimin purification and PA synthesis

Purification of pulcherrimin was modified from previously described protocols[15,19]. Briefly, B. subtilis NCIB 3610 was grown at 37 °C for 3 h in LB with agitation to reach an $OD_{600} = 1-1.5$. One mL of B. subtilis NCIB 3610 ($OD_{600}$ normalized at 1) was inoculated in 50 mL MSgg medium on a 200 mL culture flask without agitation at 37 °C for 48–72 h (until a clear red precipitate was visible). The culture was centrifuged at 4 °C

for 15 min at 27,000 × g. The supernatant was discarded, and 25 mL of NaOH 2 M was added to solubilize the red precipitate. The yellow mixture was centrifuged at 20,000 × g at 4 °C to remove precipitated iron. The clear supernatant was transferred and acidified with 36.5–38% HCl to reach pH=1. After adding HCl, the mixture started to turn red and pulcherrimin precipitation started to appear. The mixture was incubated overnight at 4 °C. Then, the mixture was centrifugated at 10,000 x g and the precipitated pulcherrimin was washed with 100% EtOH. Pulcherrimin was dried at 37 °C overnight and stored at 4 °C until needed. Details on pulcherriminic acid synthesis are provided in the supplementary information.

## β-galactosidase assays

B. subtilis biofilms were suspended in Z-buffer (40 mM $NaHPO_4$; 60 mM $Na_2HPO_4$; 1 mM $MgSO_4$; 10 mM KCl) before sonication at 30% amplitude for 30 s and $OD_{600}$ was measured. β-mercaptoethanol was added to a final concentration of 38 mM. Twenty microliter of lysozyme (20 mg/mL) was added to the bacterial suspension and incubated at 30 °C between 30 min and 1 h. All samples were diluted and 100 μL of an ONPG solution (4 mg/mL) in z-buffer with β-mercaptoethanol (38 mM) was added. When solutions started to turn yellow, 250 μL of 1 M $Na_2CO_3$ was added to stop the reaction and $OD_{420}$ and $OD_{550}$ was recorded. The miller units were calculated based on the following equation: Miller Units = 1000 x [($A_{420nm}$ −1.75 x $OD_{550nm}$)]/($T_{min}$ x $V_{mL}$ x $OD_{600}$).

## Pellicle formation and growth curves with pulcherrimin

B. subtilis cells were grown to exponential phase in LB at 37 °C for 3 h, washed, and resuspended in PBS at an $OD_{600} = 1$. Then, 15 μL were inoculated in 1 mL of MSgg in a 24-well plate. Iron chloride ($FeCl_3$) or pulcherrimin or Fe-citrate were added at 12 h post-inoculation at a final concentration of 50 μM. To evaluate growth, the content of the wells was harvested at 12 h, 18 h, 24 h, 30 h, and 36 h, sonicated to disrupt the pellicle and aggregates. The biomass was assessed by optical density at $OD_{600}$, and cells were diluted and plated on LB agar plates to count CFUs.

## In vitro dissociation kinetics assays using UPLC-MS/MS

All chemicals used in this work were analytical grade. Methanol (MeOH), water ($H_2O$), and formic acid (FA) (Optima grade for LC-MS) were purchased from Fisher Scientific (Ottawa, ON, Canada). Samples were centrifuged at 10,000 × g for 10 min at 4 °C. The supernatant was collected and cleaned by solid phase extraction (SPE) using HLB cartridges (Waters Corporation, Milford, MA). Cartridges were conditioned with 1 mL of MeOH and equilibrated with 1 mL $H_2O$. After loading 1 mL of supernatant, the SPE cartridge was rinsed with 1 mL $H_2O$ and dried for 1 min using a gentle nitrogen flow. The analyte was eluted using four times 250 μL of MeOH. Finally, the sample was filtered through a 0.22 μm PTFE syringe filter before UPLC-MS/MS analysis. Samples analyses were performed by liquid chromatography coupled with tandem mass spectrometry (UPLC, Xevo TQ MS). Compounds were separated with an HSS-T3 (100 × 2.1 mm, particle size of 1.8 μm) (Waters Corporation, Milford, MA). The mobile phase consisted of solvent A ($H_2O$ + 0.1% FA) and of solvent B (MeOH + 0.1% FA). The flow rate was set to 0.4 mL/min, and the injected volume was 3 μL. After injection, the samples were eluted with the following gradient: 0–1 min, 5% B, 1–4 min, 5–100% B, 4–6 min, 100% B, 6–7 min, 100–5% B, 7–8 min, 5% B. The efficiency of the method is shown in table S2 for linearity. The recovery and detection, and quantification limits are presented in table S3. The limits of detection (LOD) and quantification (LOQ) were determined according to the International Committee on Harmonization as three and ten times the standard error of the calibration curve[63].

## Measurement of ROS levels

B. subtilis colonies were harvested after 24 h and sonicated at 30% amplitude for 30-60 s until homogenous. The cells suspension was

then mixed with 2′,7′-dichlorodihydrofluorescin diacetate (DCFH$_2$-DA; purity ≥97%, Sigma), a fluorogenic dye that is converted to the fluorescent DCF, at a final concentration of 100 μM and incubated in the dark for 1 h. Deferoxamine mesylate (DFO; purity ≥92.5%, Sigma) and N,N,N′,N′-Tetrakis(2-pyridylmethyl)ethylenediamine (TPEN; purity ≥97%, EMD Millipore). Cells were pellets by centrifugation at 16,200 × $g$ for 1 min. The supernatant was collected (100 μL) and cells were resuspended in PBS before being were transferred into a 96-well plate (100 μL), and the fluorescent intensity was measured with a TECAN Spark monochromator-based (SparkControl version 3.1) with an excitation wavelength of 485 nm and emission wavelength of 535 nm, bandwidth 20.

### Cyclic voltammetry

To characterize the redox potential of pulcherrimin, cyclic voltammetry was performed to evaluate to what extent pulcherrimin immobilizes Fe(III) compared to known Fe chelators such as deferoxamine and EDTA. Fe-Deferoxamine et Fe-EDTA complexes were pre-formed for 1 h in the dark. For Fe-PA, the complexation was performed in the electrochemical cell 3 min prior to collecting the first voltammogram. The experiment was conducted on a CHI-1040C (CH Instruments). A 3 mm diameter glassy carbon working electrode (CHI104), a platinum plate (counter electrode), and an Ag/AgCl (KCl saturated; reference electrode) were used and acquired from CH Instruments. All voltammetric experiments were performed in deoxygenated PBS with a constant flow of argon by cycling the potential between −1.0 and 1.0 V at a scan rate of 25 mV·s$^{-1}$. All solutions were evaluated with $10^{-4}$ M of Fe (ratio 1:1; Ligand-Metal) except for pulcherriminic acid, where $2 \times 10^{-4}$ M was added (3:2 ratio; PA-Fe).

### Statistical analysis

Statistical analysis was performed with GraphPad Prism 9 version 9.4.0.

### Reporting summary

Further information on research design is available in the Nature Portfolio Reporting Summary linked to this article.

## Data availability

Source data are provided with this paper.

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

## Acknowledgements

We thank members of Beauregard laboratory, Bellenger laboratory, and Matthew F. Traxler laboratory for helpful discussions. We also thank Daniel Garneau, Philippe Venne, René Gagnon, Matthieu Fillion, and Daniel Fortin for technical advice. We thank François M. M. Morel and Anne M. Kraepiel-Morel for critical reading of the manuscript. This work was supported by a master's degree Fellowship (302246) from Fond de Recherche—Nature et Technologies to F.O, by a Doctoral fellowship (299933) from Fond de Recherche—Nature et Technologies to V.C.L, by NSERC discovery grant RGPIN-2016-03660 to J.P.B and NSERC discovery grant RGPIN-2020-07057 to P.B.B.

## Author contributions

V.C.L conceived the project. P.B.B and J.P.B supervised the development of the project. V.C.L, L.Har, M.P, and J.L acquired data. J.S.B advised and helped with strains construction. A.R, N.F, E.V.A provided technical advice. F.O and L.Hal performed all synthesis experiments. C.Y.L managed and advised synthetic chemistry. P.D.D managed and advised cyclic voltammetry experiments. V.C.L, P.B.B, and J.P.B wrote the manuscript and all authors contributed with editing and approved of the manuscript.

## Competing interests

The authors declare no competing interests.
