## [Peer Review File · Nature Communications]

Pulcherriminic acid modulates iron availability and protects against oxidative stress during microbial interactionsReviewer #1 (Remarks to the Author):

This manuscript by Charron-Lamoureux et al. investigated the role of pulcherrimin in *Bacillus subtilis*-*Pseudomonas protegens* (Pf-5) interspecies competition, cell iron management, and protection against oxidative stress. Overall, the authors presented interesting and new observations and results about the biological functions of pulcherrimin, which are not well understood beyond being an iron-sequestering molecule in *B. subtilis* and other microbes. As said, in some places, there are observations and interpretation of data that are less conclusive or convincing and can be debated. For example, the claimed biological significance and benefit of producing PA for iron management as its primary function in *B. subtilis*. In other places, additional experiments are needed to more strongly support the authors' claim. For example, how pulcherrimin prevents fenton reaction, reduces ROS production, and protects against oxidative stress. I will specify these points below.

It seems to me that the findings that PA production in *B. subtilis* is strongly induced in the presence of *P. protegens* (Pf-5) (Fig. 1b, 1d) and pulcherrimin production seems to protect *B. subtilis* from killing by Pf-5 are very interesting and results convincing (Fig. 2). The authors further tested what secreted molecules from Pf-5 induced PA production; although the answer is not entirely clear, it does seem that several secondary metabolites together (e.g. diffused antibiotic and volatile) induced PA production in *B. subtilis*. On the other hand, it is a bit surprising that the authors did not further test what killing activities from Pf-5 inhibit *B. subtilis*. This is a relatively easy test since the authors can use similar sets of Pf-5 mutants used for PA induction and known killing factors in Pf-5 are probably limited. The answer may provide important insight on how pulcherrimin protects *B. subtilis* from Pf-5.

Overall, this part of the story could lead to a very interesting *B. subtilis*-*P. protegens* interaction story with molecular details by itself. Of course, to make the story significant enough, the authors will need to provide solid evidence on what factors from Pf-5 induces PA production, how it kills *B. subtilis*, and how pulcherrimin protects *B. subtilis* against that killing factor(s).

Figure 3, these are important sets of data to support the authors' claim that PA production precipitates iron and causes iron limitation in the media, which in turn feeds back to intracellular iron homeostasis and ultimately bacillibactin production. The authors should provide quantitative assays on the activities of the *PdhbA-lacZ* reporter with or without PA production, and with or without Pf-5. Second, measuring intracellular concentrations of iron, instead of the total iron in the biofilm, will provide a cleaner picture about intracellular iron homeostasis. Third, since the *dhb* operon is known to be regulated by (at least) both iron homeostasis and developmental pathway (e.g. *Spo0A-AbrB*). It is important to show that the observed regulation by PA is mediated through iron homeostasis. Experiments such as adding excessive iron to see if it counteracts the induction may provide the answer one way or the other.

The proposed conceptual working model, although interesting, seems counterintuitive. The authors suggested that *B. subtilis* cells produce pulcherrimic acids to form insoluble pulcherrimin precipitates and thus cause local iron (Fe^{III}) stress (with and without interspecies competition). Under the self-created iron limitation, *B. subtilis* will then have to produce BB siderophore to acquire precipitated iron. First, I don't see any benefit to have this so-called pairwise mechanism. Producing BB is very costly to *B. subtilis*. This explains why *B. subtilis* produces BB only when iron in the environment is very limited. The fact that under most laboratory media conditions (Ferric iron is sufficiently provided in the media), production of BB is not needed and its loss does not impact growth at all, suggesting that production of other types of hydroxymates to bind iron and uptake iron is sufficient for cellular iron utilization in *B. subtilis*. To summarize, it will be strange to think that *B. subtilis* first self-creates iron stress by producing pulcherrimic acids and then synthesizes costly BB siderophore in order to acquire iron from the self-created stress condition. Creating such iron stress also does not seem to limit the growth of other species since lots of microbial species evolved to have the ability to acquire limited amounts of iron from the environment, especially those living in the soil. The authors also showed in this study that the growth of Pf-5 was not impacted under the hypothetical iron limitation.

About insolubility or precipitation of PA, since most free iron is precipitated in the form of various oxides complex, I don't completely understand what it means when the authors claim that PA is

secreted, binds iron, and precipitates iron in large quantities. Is it a change from various insoluble Fe-oxide complex to insoluble pulcherrimin? Could the authors explain in more details how PA production shifts the Fe bioavailability, what molecules it competes in terms of iron binding? To follow up, my personal opinion is that insolubility (or precipitation) of PA is a probably relative concept. I am not a chemist, but when comparing pulcherrimin to various iron-oxide complex, I doubt pulcherrimin (a di-peptide) is less soluble than them. The one case I can imagine is that pulcherrimic acids compete with other better soluble Fe-hydroxymate complexes (due to higher binding affinity), hijack ferric iron from them, and then form insoluble precipitates, but is there evidence for that?

The idea that PA production reduces oxidative stress and ROS accumulation is new and quite interesting. Based on what I read, the authors hypothesized that this is due to PA production decreasing environmental iron bioavailability by precipitation and reduced fenton reaction and ROS production. One thing unclear to me is that since fenton reaction mostly involves intracellular reduced ferrous iron, while most environmental irons are oxidized ferric iron. Therefore, this model will need to emphasize that PA production reduces environmental iron availability, which in turn reduces intracellular iron concentration (ferrous iron), and therefore internal ROS production. For these extra steps, more direct evidence will be needed. For example, whether the authors can detect differences in intracellular ROS accumulation (rather than the entire biofilm) between wt and yvmC, or with or without supplementation of purified pulcherrimin. If the authors are hypothesizing extracellular fenton reaction, then clear explanation will be needed in terms of how reduced ferrous iron is achieved extracellularly.

Minor point:

Line 145, there is probably no need to emphasize the impact of PA on biofilm formation, since the previous study has demonstrated otherwise, and as is this study, and the biofilm impact shown here seems to be due to indirect effect from killing by Pf-5.

Reviewer #2 (Remarks to the Author):

Metals are vital to microbial structure and function. For that Charron-Lamoureux et al describes the Fe management strategy employed by *B. subtilis* (Bsu) involving biological secondary metabolite molecules Pulcherrimin (PA, identified around 1965) and bacillibactin (BB). I praise authors for their efforts in tackling such an important question of why PA is produced? In that regard, authors put forth a model where Bsu makes PA as a way to retrieve Fe with the aid of BB for growth as well as to reduce the consequences of oxidative stress. Idea and methods used are very exciting but detailed mechanism is lacking. Often manuscript suffers from a lack of clear explanation, context for introducing/testing hypothesis, unexplainable/inconclusive data, and non-optimal figure organization. Majority of the conclusions are previously known for yeast (PA blocking biofilm, insoluble precipitation in response to Fe), Bsu PA having an antioxidant property. The new aspect would be that PA-Fe insoluble precipitates are donating Fe to BB then to Bsu. Additionally, some data are lacking controls (ctrl). Overall, these issues make it difficult to fully grasp and appreciate the strength of this manuscript. My detailed comments are as follows:

- 1) Corresponding author's contact info is missing (email)
- 2) Optional: current title of the manuscript sounds a bit kitschy, perhaps revise it.
- 3) Line 18 in the abstract needs a grammatical revision: "but which role in Fe homeostasis remains elusive" ...should read "but its role in Fe homeostasis remains elusive"
- 4) I understand the use of MSgg to promote biofilm, but use of MS is just to see visible PA synthesis while MS is not able to form a biofilm kind of makes me question how it ties in? Please provide a better and clear explanation for MS. What all key ingredient(s) might be responsible for such differences in the phenotypes among media?
- 5) Line 30 and 31 is not technically true for all the prokaryotes so maybe specify instead of generalizing it. To my knowledge Mg and K are abundant elements; further *Borrelia* can grow without iron...
- 6) Line 46-47: ...another Fe chelating molecule produced by *B. subtilis*, and other bacteria and yeast could be revised... "another Fe chelating molecule produced by yeast and many bacteria

including *B. subtilis*.

7) Line 66-67 is missing a reference...

8) Like fig 2, Fig 1a and 1b (MSgg/MS) should specify media name inside the actual fig perhaps underneath or above

9) I do not like extended data system. Fig should be either a part of main text or supplemental...

10) Line 106 is citing a wrong fig, and it should be fig 2b. Also, I suggest moving panels of fig 2a and 2b for yvmC deletion and Pf-5 yvmC panels into fig 1. This way there will be no repetitions for the ctrl panels of 3610 and Pf-5-3610. Also, with current Fig1 organization, without actual compound analysis by MS it is a stretch to use current fig1 legend title. But with addition of yvmC panels in Fig1 the title of the legend is acceptable.

11) In my opinion Panel 1d 1f, especially the way it is shown right now are adding confusion (so revised fig 1 could look like Fig 1A 1B (with inclusion of panels from fig2 A and 2B, current 1C current 1d, current fig 2C 2E...rest of the fig i.e.-1e, 1f, 2d, 2f could go to supplemental)

12) Findings of fig 1C through 1D could be articulated better, right now it's a bit untidy especially in lines 117-126.

13) Panel 2e of the fig yvmC/Pf-5 in the legend is depicted with triangle but on actual plot marker is depicted in square instead of triangle.

14) The data of fig 2e & 2f, basically suggests that Pf-5 mediated effect on yvmC deletion strain is bacteriostatic so I would suggest testing a chemical complementation. Do author's see biofilm promoting effects if Pulcherrimin is supplied exogenously? PA is not commercially available, but authors have synthesized it so it's worth testing.

15) Fig 3 data is not directly showing bacillibactin synthesis (or measured), but it shows dhbA induction, therefore legend and conclusion should indicate this indirect correlation.

16) Also, panels of fig 3C & 3D did not convince me that Pulcherrimin triggers dhbA induction. On the contrary, Pulcherrimin synthesis on MS prevents dhbA induction. It's the Pf-5 that mediates bacillibactin induction and is visible in the absence of Pulcherrimin. yvmC is showing blue color that suggests that loss of Pulcherrimin biosynthesis still leads to Bacillibactin induction. Unless there is a parallel pathway that also makes PA (which I doubt since yvmC cells are white lacking red zones whenever subjected to competition). Maybe there is a hierarchy, PA is the first line of Fe-acquisition system followed by BB (?). Although, pchR mutant is showing PA synthesis like one would expect (as described in the literature) but its dhbA induction to Pf-5 is not near comparable to yvmC deletion subjected to Pf-5 competition.

17) Line 232.. in "the" presence of

18) Fig 5 biochemical data has some poor R square values. Did author test a ctrl reaction without BB for changes of Fe bound to PA as shown in the 5a? This will indicate that this Fe transfer interaction between BB and PA is specific and not merely due to oxidation.

19) Also, Fig 5- apo to holo BB formation and holo-to apo PA turnover is taking such a long time (~72 hr) thus I question its validity in biological context.

20) Did authors observed the same effect with dhbA(B/C) mutations seen for dhbF cells? With the loss of dhbF cells are still able to make 2,3 dihydroxybenzoate a known Fe chelator, thus cytosolic enzyme functions driven by Fe might be perturbed.

21) Your cell source is not grown/coming from a highly aerobic condition (like vigorous shaking), at best it is microaerobic. How do you factor in ROS with respect to PA function?

22) It would be nice to have a benchmark of DCF signal for cell suspension grown under fully aerobic conditions and later processed with the same set of condition as 24hr biofilm. Basically, is there a difference in signal?

23) Also, DCFDA is a valid sensor of hydroxyl radicals/ferryl radicals. It does not react with superoxide or hydrogen peroxide. Lots of things can oxidize DCFDA. Metalloenzymes, Cytochrome c, purified catalase, and SOD oxidizes it. Who is the oxidant in your case? If not known, then should not be generalized as a "ROS" perhaps call it a "DCF-reactive species" or "hydroxyl/ferryl radicals".

24) Further, as an additional ctrl what happens if non-iron chelator (e.g., TPEN) is added? Do cells still retain the high DCF reactive signal?

25) It is true that when you stress microbes, you may see an elevation in DCF signal over unstressed controls. However, the DCF content of cells depends upon the energy charge, because pmf-driven efflux systems continuously pump DCF out of bacteria. So, when CCCP is added to poison the potential, a four-fold increase in DCF loading ensues. Therefore, when you load a cell with it and observe fluorescence happening due to membrane potential changes. Did author probe membrane potential across strains/conditions using DioC2 dye?

26) For Line 301, EDTA is not just a Fe chelator it can chelate many metals/it is a non-specific molecule.

27) Is it possible that the predominant job of PA is to prevent pathogen/competitor growth by Fe sequestration (similar to biocontrol activity of yeast system in generating insoluble precipitates) and for Bsu keeping BB in a soluble state? While iron release from BB and its re-uptake into Bacillus is a mere side-reaction and not the main course of action?

28) What about P. protegens that causes oxidative stress? Basically, how does B. subtilis sense the presence of P. protegens nearby? Perhaps, volatiles, but with the loss of volatiles you still see the effects? Is it just the chemistry or are there specific signal sensing mechanisms in-place for Bsu?

29) In the absence of PA synthesis, biofilm is favored on a specific media and when biofilm is formed, PA synthesis is not seen. Therefore, is there a possibility that maybe this is a cell number game? Trade-off for PA synthesis by population density (complex biofilm).

Reviewer #3 (Remarks to the Author):

Thank you for your interesting work on pulcherrimin! Please see my comments and suggestions in the attached pdf.

Reviewer #3 Attachment on the following page.

Review for “**Pulcherrimin: a bacterial Swiss army knife in the iron war**”

by Vincent Charron-Lamoureux, Lounès Haroune, Maude Pomerleau, Léo Hall, Frédéric Orban, Julie Leroux, Adrien Rizzi, Jean-Sébastien Bourassa, Nicolas Fontaine, Élodie V. d’Astous, Philippe Dauphin-Ducharme, Claude Y. Legault, Jean-Philippe Bellenger, and Pascale B. Beauregard

Summary:

In this manuscript, Charron-Lamoureux *et al.* report a new biological role of pulcherriminic acid (PA) in the bacterial competition for iron. The authors present a study based on *Bacillus subtilis* (PA producer) and *Pseudomonas protagens*, showing that *B. subtilis* secretes PA to create a local Fe³⁺ source in the form of poorly soluble pulcherrimin. Furthermore, the authors argue that precipitation of pulcherrimin prevents formation of reactive oxygen species, protecting *B. subtilis* from the deleterious effects of oxidative stress.

Overall, the manuscript meets the requirements of *Nature communications* in terms of novelty and impact. However, there are a few things that I would recommend to be addressed by the authors before accepting the manuscript for publication. Please find my questions and suggestions below.

Recommendation: revise

Comments:

1. It would be helpful if you provide the chemical structures of pulcherriminic acid and pulcherrimin in the manuscript. Also, please include the binding constants of Fe³⁺ to PA, bacillibactin, pyoverdine, DFO, EDTA, and DHB for comparison (could be added to the SI).
2. Given that there exist siderophores that bind Fe³⁺ much stronger than bacillibactin, would *B. subtilis* be able to compete for Fe³⁺ from pulcherrimin with other bacterial strains? Are there alternative explanations to the formation of PA and pulcherrimin (different from the local iron source and combating oxidative stress hypotheses)? Do you plan to test other strains to assess whether *B. subtilis* acts similarly in other co-cultures (not for this manuscript, but in future)? Would you like to elaborate a bit about potential future plans in the “Discussion” section?
3. Line 166: I think you should replace pulcherrimin with pulcherriminic acid since you’re talking about the chelator rather than the complex.

4. You've used 2',7'-dichlorofluorescein in some fluorescence assays but did not specify what where the excitation and emission wavelengths. Please make sure to add this piece of information. Also, specify which filters were used for recording the fluorescence images.

5. Lines 283-284: "As expected, DFO supplementation in the $\Delta yvmC$ mutant restored ROS levels comparable to WT in both media." – I am confused by this statement. Isn't DFO chelating Fe^{3+} and reducing the labile iron concentration, such that the concentration of ROS decreases, too? How is it that supplementation of DFO restores the ROS levels then?

6. Line 310: please reword "reduced Fe reduction".

7. The paragraph starting with line 348 is highly speculative. I do not think that you brought sufficiently valid arguments to say that "immobilizing Fe still bears merits". One could argue that if the competing microorganism is already producing a "stronger" siderophore under iron limitation, would it really cost it more energy to extract Fe^{3+} from pulcherrimin? In fact, *B. subtilis* seems to be at loss as it has to make both PA and BB to obtain ferric iron. The xenosiderophores argument and cross-feeding can go both ways as well.

Also, please be careful with wording in lines 356-357 and be reminded that kinetic lability/inertness is not the same thing as thermodynamic stability. Overall, I'd suggest rewriting this paragraph as it looks somewhat weak in terms of arguing that the secretion of PA by *B. subtilis* and formation of pulcherrimin will cost the competitor strain more energy.

8. In the contributions section, line 702: none of the authors have their initials as "F. B." Please correct.

9. The "Methods" section: please provide the centrifugation parameters as factors of the g -force rather than RPMs. For example, in the "Flow cytometry" and "Measurement of ROS levels" you give RPMs, whereas in the rest of the sections you use " $x g$ ".

10. For the information on synthesis (SI), please add the details regarding the type of the TLC plates used, silica gel (mesh etc.), the commercial sources for the chemicals/solvents, the apparatus used for m.p. determination, NMR and MS instrumentation.

11. Is there a reason why HRMS is only reported for 2 molecules? Also, could you please add C-13 NMR for pulcherriminic acid, too?

The following questions/suggestions are all pertaining to the SI:

12. Synthesis of diketopiperazine (1): may you please add the volumes of AcOEt and isopropyl alcohol in line 10?

13. Synthesis of molecules (2) and (3): on the reaction scheme you give 5h as reaction time, whereas the procedure states 2h. So which is it? Please correct.

14. R_f (1% ether in hexanes) of molecule (2) is 0.05. How did you extract the compound from silica gel? Please specify if you used another solvent/solvent mixture to remove (2) from the column after eluting (3) with 1% ether in hexanes.

15. Please check and correct the number of moles of (2) and volume of DCM in line 45. I believe it should be 829 micromol and 1.66 mL, respectively, but please double-check and amend.

16. Line 71: what solvent was sodium methoxide dissolved in?

17. Line 79: replace pulcherrimin by pulcherriminic acid.

18. Pages 5-9: please correct the “cipher” of the molecules given in parenthesis on the NMR spectra, so that they match the numbers provided in the synthetic procedures. Also, please correct the name of molecule (5) given in parenthesis on page 9: it should be pulcherriminic acid instead of pulcherrimin.

19. Table S2: please translate the title of the third column into English.

20. For table S3, how were the LOD and LOQ determined? Would be good to add this piece of information to the methods section.

21. Lastly, some specific questions and suggestions regarding the NMR spectra:

(a) Please add the integration values onto the H-1 spectrum of molecule (1) on page 5.

(b) For 3-chloro-2,5-diisobutylpyrazine: the signals at 2.21 and 2.10 ppm do not look like dp to me (also, it's more commonly called quintet rather than pentet). Is the CH proton of the isobutyl group supposed to be a dp at all? Also, I do not agree with the dd assignment for the 0.95 ppm signal. Isn't it corresponding to the two overlapping doublets of the four methyl groups?

(c) In the spectra of 2,5-dichloro-3,6-diisobutylpyrazine, 2,5-dichloro-3,6-diisobutylpyrazine 1,4-dioxide and 2,5-dihydroxy-3,6-diisobutylpyrazine 1,4-dioxide, the signals labeled as dp should be relabeled as multiplets.

As a general recommendation for the spectra: please double check all splitting pattern assignment, J -values and integrals. Also, it would be more illustrative and easier for the reader if you zoomed in the multiplet signals and included them as insets into the spectra.

REVIEWER COMMENTS

Reviewer #1 (Remarks to the Author):

This manuscript by Charron-Lamoureux et al. investigated the role of pulcherrimin in *Bacillus subtilis*-*Pseudomonas protegens* (Pf-5) interspecies competition, cell iron management, and protection against oxidative stress. Overall, the authors presented interesting and new observations and results about the biological functions of pulcherrimin, which are not well understood beyond being an iron-sequestering molecule in *B. subtilis* and other microbes. As said, in some places, there are observations and interpretation of data that are less conclusive or convincing and can be debated. For example, the claimed biological significance and benefit of producing PA for iron management as its primary function in *B. subtilis*. In other places, additional experiments are needed to more strongly support the authors' claim. For example, how pulcherrimin prevents fenton reaction, reduces ROS production, and protects against oxidative stress. I will specify these points below.

It seems to me that the findings that PA production in *B. subtilis* is strongly induced in the presence of *P. protegens* (Pf-5) (Fig. 1b, 1d) and pulcherrimin production seems to protect *B. subtilis* from killing by Pf-5 are very interesting and results convincing (Fig. 2). The authors further tested what secreted molecules from Pf-5 induced PA production; although the answer is not entirely clear, it does seem that several secondary metabolites together (e.g. diffused antibiotic and volatile) induced PA production in *B. subtilis*. On the other hand, it is a bit surprising that the authors did not further test what killing activities from Pf-5 inhibit *B. subtilis*. This is a relatively easy test since the authors can use similar sets of Pf-5 mutants used for PA induction and known killing factors in Pf-5 are probably limited. The answer may provide important insight on how pulcherrimin protects *B. subtilis* from Pf-5.

Overall, this part of the story could lead to a very interesting *B. subtilis*-*P. protegens* interaction story with molecular details by itself. Of course, to make the story significant enough, the authors will need to provide solid evidence on what factors from Pf-5 induces PA production, how it kills *B. subtilis*, and how pulcherrimin protects *B. subtilis* against that killing factor(s).

We agree with the reviewer that exploring the killing factors could be very interesting. Similarly, to the environmental cue(s) triggering the production of pulcherrimin, we expect the killing factor(s) to be multifactorial, which would make their identification much more difficult. While this mechanism is of foremost interest, what is proposed here by the reviewer, i.e., what factors from Pf-5 induce PA production, how it kills and how pulcherrimin protects *B. subtilis*, goes well beyond the scope of our current article and is destined to be the entire MSc project of a student who just started.

Figure 3, these are important sets of data to support the authors' claim that PA production precipitates iron and causes iron limitation in the media, which in turn feeds back to intracellular iron homeostasis and ultimately bacillibactin production. The authors should provide quantitative assays on the activities of the *P_{dhbA-lacZ}* reporter with or without PA production, and with or without Pf-5.

We agree with the reviewer that the qualitative pictures showing *P_{dhbA-lacZ}* reporter are insufficient, and we performed the quantification of these assays at specific time points (MS; 24 h and MSgg 48 h). We clearly show in Fig. 3e,f that the expression of *P_{dhbA-lacZ}* is more induced next to Pf-5 and when PA is produced. Line 183.

Second, measuring intracellular concentrations of iron, instead of the total iron in the biofilm, will provide a cleaner picture about intracellular iron homeostasis.

We performed the experiment as suggested by the reviewer (Fig. S3). Line 604. Briefly, the intracellular concentration of Fe remains the same between all conditions in MS medium where Fe is provided as the organic complex (Fe-EDTA). On MSgg (FeCl₃), we observed that the *yvmC*-Pf-5 interaction leads to higher intracellular Fe concentration compared to all conditions tested. It is possible that in absence of pulcherriminic acid production, the strategy of the *yvmC* mutant (PA non-producer) shift to luxury storage of intracellular Fe, a well-known nutrient management strategy when facing competition for valuable resources (Mazancourt and Schwartz, 2012, Starve a competitor: evolution of luxury consumption as a competitive strategy).

Third, since the *dhb* operon is known to regulated by (at least) both iron homeostasis and developmental pathway (e.g. Spo0A-AbrB). It is important to show that the observed regulation by PA is mediated through iron homeostasis. Experiments such as adding excessive iron to see if it counteracts the induction may provide the answer one way or the other.

We performed the experiment as suggested by the reviewer. We added 5x more Fe-EDTA in MS to see if it stops the expression of $P_{dhbA-lacZ}$ (Fig. S4c). Line 610. Briefly, although the miller units follow a similar pattern between 1x Fe-EDTA and 5x EDTA, it delayed the expression of $P_{dhbA-lacZ}$. It took more time (see data source) to detect $P_{dhbA-lacZ}$ in the Beta-galactosidase assay. These results suggest that in presence of Fe-EDTA excess, it probably take more time for PA to depleted Fe in the media which delay $P_{dhbA-lacZ}$ expression.

The proposed conceptual working model, although interesting, seems counterintuitive. The authors suggested that *B. subtilis* cells produce pulcherrimic acids to form insoluble pulcherrimin precipitates and thus cause local iron (FeIII) stress (with and without interspecies competition). Under the self-created iron limitation, *B. subtilis* will then have to produce BB siderophore to acquire precipitated iron. First, I don't see any benefit to have this so-called pairwise Producing BB is very costly to *B. subtilis*. This explains why *B. subtilis* produces BB only when iron in the environment is very limited. The fact that under most laboratory media conditions (Ferric iron is sufficiently provided in the media), production of BB is not needed, and its loss does not impact growth at all, suggesting that production of other types of hydroxymates to bind iron and uptake iron is sufficient for cellular iron utilization in *B. subtilis*.

The control of Fe in growth culture is tricky. Fe is provided to growth media either as salt (e.g., $FeCl_3$) or as an organic complex (Fe-EDTA). In both cases, Fe is not readily available to bacteria. Ferric Fe released in the oxic medium, precipitates to form ferric hydroxides and oxides. As for Fe-EDTA, it is not bioavailable. Thus, when salt or Fe-EDTA are used, free Fe in the solution is extremely low. In both cases microorganisms rely on their active acquisition systems (e.g., siderophores) to retrieve Fe from hydroxides/oxides or the Fe-EDTA complex (either directly, or by favouring Fe release from the Fe-EDTA following depletion of free Fe in the solution). The importance of siderophores for Fe acquisition in laboratory cultures is well documented, in many bacteria. As for *B. subtilis* in MSgg, siderophores are required to support Fe homeostasis. According to Rizzi *et al.* 2019 (Fig. S5; <https://journals.asm.org/doi/full/10.1128/AEM.02439-18>), both BB and DHB are produced in MSgg containing 100 μM $FeCl_3$ which is twice the amount of Fe used in our study (we use MSgg at 50 μM $FeCl_3$). Importantly, *B. subtilis* NCIB 3610 does not produce other hydroxamate siderophore and relies only on DHB (precursor of bacillibactin) and bacillibactin (BB) to grow. Thus, the production of pulcherriminic acid (and precipitation of Fe as pulcherrimin complex) does not fundamentally change the challenge bacteria face, as Fe is present in the medium under chemical forms that are not readily available for uptake (in oxic conditions). What is significantly affected by the production of pulcherrimin is the chemical form of that Fe pool. We discuss in length in the discussion the benefices that precipitating Fe as a pulcherrimin complex could provide for the control of Fe acquisition, especially in real soil. We also discuss the question of the cost of pulcherrimin and siderophores production, especially the benefits that actively storing Fe close to the cells could provide the context of interspecies competition considering other features of *Bacillus*, such as its known ability to use xenosiderophores.

To summarize, it will be strange to think that *B. subtilis* first self-creates iron stress by producing pulcherrimic acids and then synthesize costly BB siderophore in order to acquire iron from the self-created stress condition. Creating such iron stress also does not seem to limit the growth of other species since lots of microbial species evolved to have the ability to acquire limited amounts of iron from the environment, especially those living in the soil. The authors also showed in this study that the growth of Pf-5 was not impacted under the hypothetical iron limitation.

We agree with the reviewer that our observations are, at least at first glance, counterintuitive. We clearly state it in the introduction and the abstract as it is indeed one of the most fascinating findings of this study. However, limiting the interpretation solely to pulcherrimin and siderophore production cost and pulcherrimin efficiency to forfeit competitor ability to retrieve Fe would be too reductionist. The field of Fe homeostasis is an exciting and fast-evolving research field. For instance, we do not clearly grasp the real cost of Fe assisted acquisition as it cannot be simply estimated based on the sole cost of siderophore production but by assessing siderophore use efficiency (meaning the amount of Fe retrieved related to the amount of siderophore produced, which in real environments depends in part on siderophore loss). While kinetics is rarely taken into account, the retrieval of Fe from oxides or EDTA is strongly kinetic limited. Thus, the efficiency of a given siderophore to retrieve Fe does not simply rely on its own affinity for Fe but its relative affinity (compared to the Fe source) as well as the kinetic of the reaction. As for the ability of pulcherrimin to prevent competitors from acquiring Fe, we agree that our data clearly show it does not. However, it is highly unlikely that any organism can develop a strategy based on manipulation of intracellular Fe chemistry that totally inhibits Fe acquisition by competitors considering the high diversity of siderophores, and other Fe acquisition strategies, most organisms own as well as the ability of many bacteria to hijack xenosiderophores. However, using the right tool to harden Fe acquisition by competitors, even if not totally forfeiting it, can provide huge benefices, especially in real life environment.

We provide, in the discussion section, how these counterintuitive observations could indeed provide benefit to *Bacillus* considering current knowledge in the larger context of siderophore assisted acquisition of Fe and Fe homeostasis.

About insolubility or precipitation of PA, since most free iron is precipitated in the form of various oxides complex, I don't completely understand what it means when the authors claim that PA is secreted, binds iron, and precipitates iron in large quantities. Is it a change from various insoluble Fe-oxide complex to insoluble pulcherrimin? Could the authors explain in more details how PA production shifts the Fe bioavailability, what molecules it competes in terms of iron binding?

We explained in more details how PA shifts Fe bioavailability. Lines 98-102. As mentioned above, in both media: MSgg (Fe salt; insoluble Fe) and MS (Fe-EDTA; soluble Fe), Fe is not readily available for uptake. Thus, the precipitation of Fe as a Fe-pulcherrimin complex does not fundamentally change the challenge the bacteria face. In all cases readily available Fe is very low. However, it significantly changes the chemical form of Fe from which the bacteria as to retrieve Fe. As for the competition with other Fe forms (Fe hydroxide and Fe-EDTA), we observed that pulcherrimin has a high affinity for Fe and the complexation is fast. While we did not perform competition experiments, our data clearly show that the characteristic red Fe-pulcherrimin complex is formed under our experimental conditions, when Fe is provided either as Fe salt or Fe-EDTA.

To follow up, my personal opinion is that insolubility (or precipitation) of PA is a probably relative concept. I am not a chemist, but when comparing pulcherrimin to various iron-oxide complex, I doubt pulcherrimin (a di-peptide) is less soluble than them. The one case I can imagine is that pulcherrimic acids compete with other better soluble Fe-hydroxymate complexes (due to higher binding affinity), hijack ferric iron from them, and then form insoluble precipitates, but is there evidence for that?

Pulcherrimic acid is definitively soluble, but pulcherrimin is not. Upon binding to Fe, pulcherrimic acid form a lattice with the Fe^{3+} (pulcherrimic acid has 4 sites for complexation and Fe^{3+} needs 6 sites) as described in (Kluyver *et al.* 1953; see the chemical structure below). Those structure are highly insoluble, and insolubility of pulcherrimin was reported in Melvydas *et al.* 2016. We also observed immediate precipitation of pulcherrimin upon addition of $FeCl_3$ to PA, as shown in Fig 5c.

FIGURE 7

Pulcherrimin: unit configuration

FIGURE 8

Pulcherrimin: molecular pattern

However, we did not directly compare the solubility of Fe-pulcherrimin with the solubility of Fe hydroxide and oxide form in the medium (which would be very challenging). The pulcherrimin complex has a characteristic red color, which was used to observe complex formation in our experiments. As reported in the manuscript the red complex is formed illustrating that pulcherrimin efficiently retrieved Fe in presence of the Fe sources provided to the medium. We

are not sure to fully understand the second part of the reviewer's comment regarding hydroxamates and solubility. The solubility of the complex and the affinity of the ligand for Fe are distinct properties and are not directly related. Also, *Bacillus* does not produce hydroxamate siderophores. It is well documented that precipitation can hinder Fe retrieval by siderophores. However, we show on Fig 5 that Fe can be retrieved from Fe-PA by bacillibactin.

The idea that PA production reduces oxidative stress and ROS accumulation is new and quite interesting. Based on what I read, the authors hypothesized that this is due to PA production decreasing environmental iron bioavailability by precipitation and reduced fenton reaction and ROS production. One thing unclear to me is that since fenton reaction mostly involves intracellular reduced ferrous iron, while most environmental irons are oxidized ferric iron. Therefore, this model will need to emphasize that PA production reduces environmental iron availability, which in turn reduces intracellular iron concentration (ferrous iron), and therefore internal ROS production. For these extra steps, more direct evidence will be needed. For example, whether the authors can detect differences in intracellular ROS accumulation (rather than the entire biofilm) between wt and yvmC, or with or without supplementation of purified pulcherrimin. If the authors is hypothesizing extracellular fenton reaction, then clear explanation will be needed in terms of how reduced ferrous iron is achieved extracellularly.

Extracellular Electron Transfer (EET) have been shown in *B. subtilis* biofilm in MSgg where extracellular Fe(III) is reduced to Fe(II). This is stated in the discussion (Qin *et al.* 2019 <https://www.nature.com/articles/s41467-019-11681-0#Sec2>). Lines 404-405. Thus, the Fenton reaction can occur in the biofilm and form ROS contributing to the oxidative stress. Thus, our hypothesis is that oxidative stress arising from extracellular Fenton reaction (among others) would be controlled by pulcherrimin. Whether or not extracellular precipitation of Fe by pulcherrimin could also contribute to affect ROS formation inside the cell is interesting. We quantified the internal ROS production and it follow the same trend detected extracellularly (see Supplementary Figure S8).

Minor point:

Line 145, there is probably no need to emphasize the impact of PA on biofilm formation, since the previous study has demonstrated otherwise, and as is this study, and the biofilm impact shown here seems to be due to indirect affect from killing by Pf-5.

We agree with the reviewer, and we removed the sentence.

Reviewer #2 (Remarks to the Author):

Metals are vital to microbial structure and function. For that Charron-Lamoureux et al describes the Fe management strategy employed by *B. subtilis* (Bsu) involving biological secondary metabolite molecules Pulcherrimin (PA, identified around 1965) and bacillibactin (BB). I praise authors for their efforts in tackling such an important question of why PA is produced? In that regard, authors put forth a model where Bsu makes PA as a way to retrieve Fe with the aid of BB for growth as well as to reduce the consequences of oxidative stress. Idea and methods used are very exciting but detailed mechanism is lacking. Often manuscript suffers from a lack of clear explanation, context for introducing/testing hypothesis, unexplainable/inconclusive data, and non-optimal figure organization. Majority of the conclusions are previously known for yeast (PA blocking biofilm, insoluble precipitation in response to Fe), Bsu PA having an antioxidant property. The new aspect would be that PA-Fe insoluble precipitates are donating Fe to BB then to Bsu. Additionally, some data are lacking controls (ctrl). Overall, these issues make it difficult to fully grasp and appreciate the strength of this manuscript. My detailed comments are as follows:

1) Corresponding author's contact info is missing (email)

We added the missing information. Lines 15-16.

2) Optional: current title of the manuscript sounds a bit kitschy, perhaps revise it.

We agree with the reviewer, and we changed the title to: Unraveling the counterintuitive Fe management strategy mediated by pulcherriminic acid in microbial interactions

3) Line 18 in the abstract needs a grammatical revision: "but which role in Fe homeostasis remains elusive" ...should read "but its role in Fe homeostasis remains elusive"

Mistake was fixed. Line 20.

4) I understand the use of MSgg to promote biofilm, but use of MS is just to see visible PA synthesis while MS is not able to form a biofilm kind of makes me question how it ties in? Please provide a better and clear explanation for MS. What all key ingredient(s) might be responsible for such differences in the phenotypes among media?

Briefly, MS contains ammonium nitrate, potassium nitrate which are not present in MSgg. Glycerol and glutamate are supplemented in MS to allow biofilm formation, and the Fe source is also different. We indicated this in the result section. Lines 97-98.

5) Line 30 and 31 is not technically true for all the prokaryotes so maybe specify instead of generalizing it. To my knowledge Mg and K are abundant elements; further *Borrelia* can grow without iron...

Yes, we agree, and the text has been modified accordingly. Line 33.

6) Line 46-47: ...another Fe chelating molecule produced by *B. subtilis*, and other bacteria and yeast could be revised... "another Fe chelating molecule produced by yeast and many bacteria including *B. subtilis*."

We agree and the sentence has been changed. Line 49.

7) Line 66-67 is missing a reference...

References were added. Line 69.

8) Like fig 2, Fig 1a and 1b (MSgg/MS) should specify media name inside the actual fig perhaps underneath or above

This was fixed.

9) I do not like extended data system. Fig should be either a part of main text or supplemental...

The figures have been transferred from extended to supplementary data as suggested.

10) Line 106 is citing a wrong fig, and it should be fig 2b. Also, I suggest moving panels of fig 2a and 2b for yvmC deletion and Pf-5 yvmC panels into fig 1. This way there will be no repetitions for the ctrl panels of 3610 and Pf-5-3610. Also, with current Fig1 organization, without actual compound analysis by MS it is a stretch to use current fig1 legend title. But with addition of yvmC panels in Fig1 the title of the legend is acceptable.

The organization of figures has been changed.

11) In my opinion Panel 1d 1f, especially the way it is shown right now are adding confusion (so revised fig 1 could look like Fig 1A 1B (with inclusion of panels from fig2 A and 2B, current 1C current 1d, current fig 2C 2E...rest of the fig i.e.-1e, 1f, 2d, 2f could go to supplemental)

We agree with the reviewer and changes in figures organization have been made.

12) Findings of fig 1C through 1D could be articulated better, right now it's a bit untidy especially in lines 117-126.

This section will be rewritten to clarify. Lines 115-120.

13) Panel 2e of the fig yvmC/Pf-5 in the legend is depicted with triangle but on actual plot marker is depicted in square instead of triangle.

This error has been fixed. Thanks for catching this mistake.

14) The data of fig 2e & 2f, basically suggests that Pf-5 mediated effect on yvmC deletion strain is bacteriostatic so I would suggest testing a chemical complementation. Do author's see biofilm promoting effects if Pulcherrimin is supplied exogenously? PA is not commercially available, but authors have synthesized it so it's worth testing.

Synthesizing PA is highly time and resource consuming (over 4 months), and we only have 10 mg left. Considering the limited amount of compound available we are using it parsimoniously. The proposed experiment, testing biofilm induction by pulcherrimin, would need us to synthesize more compound (>30 mg), asking 4 months of our collaborator's work, and we do not consider it adds significant information to the study.

15) Fig 3 data is not directly showing bacillibactin synthesis (or measured), but it shows dhbA induction, therefore legend and conclusion should indicate this indirect correlation.

Sentences has been rewritten to remove confusion.

16) Also, panels of fig 3C & 3D did not convince me that Pulcherrimin triggers dhbA induction. On the contrary, Pulcherrimin synthesis on MS prevents dhbA induction. It's the Pf-5 that mediates bacillibactin induction and is visible in the absence of Pulcherrimin. yvmC is showing blue color that suggests that loss of Pulcherrimin biosynthesis still leads to Bacillibactin induction. Unless there is a parallel pathway that also makes PA (which I doubt since yvmC cells are white lacking red zones whenever subjected to competition). Maybe there is a hierarchy, PA is the first line of Fe-acquisition system followed by BB (?). Although, pchR mutant is showing PA synthesis like one would expect (as described in the literature) but its dhbA induction to Pf-5 is not near comparable to yvmC deletion subjected to Pf-5 competition.

We agree that these figures are not clear, and some of this contradiction stems from the fact that the blue color is not visible when overlaid by a significant amount of pulcherrimin. Pf-5 – 3610 on MS is actually very blue, displaying strong dhb activity. We performed the quantification of $P_{dhbA-lacZ}$, which solve most of the questions here (see Fig. 3e,f). Briefly, the expression of $P_{dhbA-lacZ}$ is more induced when PA is produced. We also modified the text accordingly. We agree that our result indicates a hierarchy as suggested by the reviewer.

17) Line 232 in "the" presence of

The mistake has been corrected. Line 249.

18) Fig 5 biochemical data has some poor R square values. Did author test a ctrl reaction without BB for changes of Fe bound to PA as shown in the 5a? This will indicate that this Fe transfer interaction between BB and PA is specific and not merely due to oxidation.

We added data (Supplementary Figure S7a,b) showing that the mass of the pulcherrimin precipitate does not significantly changes from 0 h to 96 h incubation without BB; we also added data showing the stability of PA (soluble fraction) through time; both of these controls validate that pulcherrimin and PA is not oxidized through time.

19) Also, Fig 5- apo to holo BB formation and holo-to apo PA turnover is taking such a long time (~72 hr) thus I question its validity in biological context.

Multiple evidence in the literature showed comparable dissociation kinetics from precipitated Fe. See: Wichard *et al.* 2009; Fig 5 (<https://pubs.acs.org/doi/10.1021/es8037214>), Coccozza *et al.* 2002; Fig 2 (<https://www.sciencedirect.com/science/article/abs/pii/S0016703701007803>). As for the validity in biological context we do not think this slow kinetics invalidate the benefice of PA-Fe precipitation. While we are used to work with fast growing bacteria in laboratory setting, in real environment growth are significantly slower. Also as mentioned above, slow kinetics in Fe-siderophore complex formation (transcomplexation) is common. So siderophore assisted acquisition of Fe faces similar kinetic challenges with Fe oxides, Fe-organic complexes and PA-Fe. We would even argue that precipitating Fe close to the cell (in the biofilm) present clear benefice because of this slow kinetics. It might help reducing the loss of siderophores which is due to diffusion away from the cell and oxidation. Actively creating a stock of Fe in a controlled environment (the biofilm) could address, at least in part, these challenges.

20) Did authors observed the same effect with dhbA(B/C) mutations seen for dhbF cells? With the loss of dhbF cells are still able to make 2,3 dihydroxybenzoate a known Fe chelator, thus cytosolic enzyme functions driven by Fe might be perturbed.

The experiment, suggested by the reviewer, has been performed and no growth has been observed for the *dhbA-F* with pulcherrimin has Fe source (Supplementary Figure 5).

21) Your cell source is not grown/coming from a highly aerobic condition (like vigorous shaking), at best it is microaerobic. How do you factor in ROS with respect to PA function?

Biofilm formation on MSgg and MS are done aerobically. Although it exists an oxygen gradient in the biofilm and the bottom layers of the biofilm lack oxygen. However recent study on Extracellular Electron Transfer (EET) in *B. subtilis* biofilm in MSgg reported that Fe(III) is reduced to Fe(II) showing that redox reactions occur in the biofilm (Qin et al. 2019).

22) It would be nice to have a benchmark of DCF signal for cell suspension grown under fully aerobic conditions and later processed with the same set of condition as 24hr biofilm. Basically, is there a difference in signal?

We performed the experiment and added the data in Supplementary Figure S8. Cells were cultivated for 3 h in shaking condition in LB and signal for ROS is low.

23) Also, DCFDA is a valid sensor of hydroxyl radicals/ferryl radicals. It does not react with superoxide or hydrogen peroxide. Lots of things can oxidize DCFDA. Metalloenzymes, Cytochrome c, purified catalase, and SOD oxidizes it. Who is the oxidant in your case? If not known, then should not be generalized as a "ROS" perhaps call it a "DCF-reactive species" or "hydroxyl/ferryl radicals".

We did some research on DCFH2-DA probe, and according to Murphy *et al.*, 2022 *Nature metabolism*, it appears to be a general oxidative stress fluorescent probe and consequently "ROS" is used in that article. Indeed, it does not react with H₂O₂ but only when this latter is converted to more reactive species in presence of redox-active metals. We include that detail at line 299, and thus we think it is justified to use the term ROS as a non-specific probe in our case. However, if the reviewer is not convinced by our argument, we will change the term ROS to hydroxyl/ferryl radicals.

24) Further, as an additional ctrl what happens if non-iron chelator (e.g., TPEN) is added? Do cells still retain the high DCF reactive signal?

We performed the experiment has suggested by the reviewer, see supplementary Figure S8. Briefly, the extracellular oxidative stress is mitigated similarly by DFO and TPEN, but it seems that DFO is better at reducing intracellular oxidative stress than TPEN. Indeed, TPEN has been shown to have high affinity toward Zn but still able to bind other metals (e.g., Fe; see Blindauer *et al.*, 2006. *Polyhedron*).

25) It is true that when you stress microbes, you may see an elevation in DCF signal over unstressed controls. However, the DCF content of cells depends upon the energy charge, because pmf-driven efflux systems continuously pump DCF out of bacteria. So, when CCCP is added to poison the potential, a four-fold increase in DCF loading ensues. Therefore, when you load a cell with it and observe fluorescence happening due to membrane potential changes. Did author probe membrane potential across strains/conditions using DioC2 dye?

Importantly, the DCF fluorescence we observed showed the same tendency for extracellular and intracellular for both media (see Supplementary Figure S8), strongly suggesting that membrane potential does not have an impact on our observation.

26) For Line 301, EDTA is not just a Fe chelator it can chelate many metals/it is a non-specific molecule.

We changed Fe for metals. Line 331.

27) Is it possible that the predominant job of PA is to prevent pathogen/competitor growth by Fe sequestration (similar to biocontrol activity of yeast system in generating insoluble precipitates) and for Bsu keeping BB in a soluble state? While iron release from BB and its re-uptake into Bacillus is a mere side-reaction and not the main course of action?

Yes, it could be it; however, in this very complex system and because we do not know the exact kinetic of each reaction, it is difficult to define for sure if BB-mediated reaction are side-reaction or truly essential for *B. subtilis* growth.

28) What about *P. protegens* that causes oxidative stress? Basically, how does *B. subtilis* sense the presence of *P. protegens* nearby? Perhaps, volatiles, but with the loss of volatiles you still see the effects? Is it just the chemistry or are there specific signal sensing mechanisms in-place for *Bsu*?

This is an excellent question, for which we do not have an answer yet, except that it is likely multifactorial. However, this question underlies the MSc project of a student who just started, and we hope to be able to provide more answer in the coming year.

29) In the absence of PA synthesis, biofilm is favored on a specific media and when biofilm is formed, PA synthesis is not seen. Therefore, is there a possibility that maybe this is a cell number game? Trade-off for PA synthesis by population density (complex biofilm).

The interplay between PA production and biofilm are complex, and we do not have yet a clear picture of it. In MSgg, PA is produced concomitantly to biofilm formation so in this case, the more cells, the more PA production. In MS, both WT and *yvmC* colonies are wrinkled and do not produce PA. Consequently, we think that PA production is linked to a specific environmental cue that is not necessarily associated with biofilm or cell density; we will investigate this question at the same time that we will look at the Pf-5 signal triggering PA production.

Reviewer #3 (Remarks to the Author):

Comments:

1. It would be helpful if you provide the chemical structures of pulcherriminic acid and pulcherrimin in the manuscript. Also, please include the binding constants of Fe³⁺ to PA, bacillibactin, pyoverdine, DFO, EDTA, and DHB for comparison (could be added to the SI).

Structures and known binding constant have been added; See Supplementary Figure S6. Binding constants of all chelators have been added, except the one of PA since it is unknown. While knowing PA binding constant could be helpful, it is a compound consuming and challenging task as the complex precipitates. However, we provide competition experiment that experimentally show that BB can retrieve Fe from the PA-Fe complex.

2. Given that there exist siderophores that bind Fe³⁺ much stronger than bacillibactin, would *B. subtilis* be able to compete for Fe³⁺ from pulcherrimin with other bacterial strains? Are there alternative explanations to the formation of PA and pulcherrimin (different from the local iron source and combating oxidative stress hypotheses)? Do you plan to test other strains to assess whether *B. subtilis* acts similarly in other co-cultures (not for this manuscript, but in future)? Would you like to elaborate a bit about potential future plans in the "Discussion" section?

Yes, future plans have added in the section discussion; see lines 358-3361. We do intent to examine how production of pulcherrimin affect the fitness of *B. subtilis* in presence of other bacterial strains and plant host. For now, we do not have other explanation for the formation of pulcherrimin, but further investigation might reveal other roles. We will add more information about these plans in the discussion.

3. Line 166: I think you should replace pulcherrimin with pulcherriminic acid since you're talking about the chelator rather than the complex.

The mistake has been corrected. Thanks for pointing that error. Line 167.

4. You've used 2',7'-dichlorofluorescein in some fluorescence assays but did not specify what where the excitation and emission wavelengths. Please make sure to add this piece of information. Also, specify which filters were used for recording the fluorescence images.

The excitation and emission wavelengths of 2',7'-dichlorofluorescein was added in the methods section under measurements of ROS levels. Lines 556-557. For the fluorescence images the camera DFC3000 G (grayscale) was used, and this information is found in the methods section under Microscopy. Lines 461-462.

5. Lines 283-284: "As expected, DFO supplementation in the $\Delta yvmC$ mutant restored ROS levels comparable to WT in both media." – I am confused by this statement. Isn't DFO chelating Fe³⁺ and reducing the labile iron concentration, such that the concentration of ROS decreases, too? How is it that supplementation of DFO restores the ROS levels then?

Indeed, our sentence is badly formulated and has been rewritten. Line 298-299.

6. Line 310: please reword "reduced Fe reduction".

We fixed the sentence. Line 307-312.

7. The paragraph starting with line 348 is highly speculative. I do not think that you brought sufficiently valid arguments to say that "immobilizing Fe still bears merits". One could argue that if the competing microorganism is already producing a "stronger" siderophore under iron limitation, would it really cost it more energy to extract Fe³⁺ from pulcherrimin? In fact, *B. subtilis* seems to be at loss as it has to make both PA and BB to obtain ferric iron. The xenosiderophores argument and cross-feeding can go both ways as well. Also, please be careful with wording in lines 356-357 and be reminded that kinetic lability/inertness is not the same thing as thermodynamic stability. Overall, I'd suggest rewriting this paragraph as it looks somewhat weak in terms of arguing that the secretion of PA by *B. subtilis* and formation of pulcherrimin will cost the competitor strain more energy.

We agree. This section was clumsy. The paragraph has been rewritten. Lines 378-388.

8. In the contributions section, line 702: none of the authors have their initials as "F. B." Please correct.

Mistake has been fixed.

9. The "Methods" section: please provide the centrifugation parameters as factors of the g-force rather than RPMs. For example, in the "Flow cytometry" and "Measurement of ROS levels" you give RPMs, whereas in the rest of the sections you use "x g".

We change RPM to x g to be constant across the manuscript.

10. For the information on synthesis (SI), please add the details regarding the type of the TLC plates used, silica gel (mesh etc.), the commercial sources for the chemicals/solvents, the apparatus used for m.p. determination, NMR and MS instrumentation.

We have added a "Pulcherriminic Acid Synthesis - General Details" to provide this additional information.

11. Is there a reason why HRMS is only reported for 2 molecules? Also, could you please add C-13 NMR for pulcherriminic acid, too? The following questions/suggestions are all pertaining to the SI:

Compounds 1 and 2 were reported in the literature and their NMR analyses were consistent with the reported data, so we considered their HRMS to be unimportant. The omission of the HRMS and ¹³C NMR of pulcherriminic acid was however a mistake, we added the information, as well as the IR, to the revised SI. All new compounds now have HRMS.

12. Synthesis of diketopiperazine (1): may you please add the volumes of AcOEt and isopropyl alcohol in line 10? The volumes were added to the procedure.

13. Synthesis of molecules (2) and (3): on the reaction scheme you give 5h as reaction time, whereas the procedure states 2h. So which is it? Please correct.

We have corrected the value, it is indeed 2 h reaction time.

14. R_f (1% ether in hexanes) of molecule (2) is 0.05. How did you extract the compound from silica gel? Please specify if you used another solvent/solvent mixture to remove (2) from the column after eluting (3) with 1% ether in hexanes.

We have verified and we confirm this is the correct R_f for molecule (2). During the purification, we use a slight gradient increase toward the end (1% to 2% Et₂O / hexanes) to elute molecule (2). We have modified the procedure to reflect this gradient.

15. Please check and correct the number of moles of (2) and volume of DCM in line 45. I believe it should be 829 micromol and 1.66 mL, respectively, but please double-check and amend.

We have corrected the values. Thank you for catching these errors.

16. Line 71: what solvent was sodium methoxide dissolved in?

Sodium methoxide was weighted in a glove box and added as a solid in the sealed tube used for the reaction. Intermediate 4 was then added, and finally 1,4-dioxane was added to the sealed tube to solubilize these two solids. We have changed the procedure to reflect this more clearly.

17. Line 79: replace pulcherrimin by pulcherriminic acid.

We have made the correction.

18. Pages 5-9: please correct the "cipher" of the molecules given in parenthesis on the NMR spectra, so that they match the numbers provided in the synthetic procedures. Also, please correct the name of molecule (5) given in parenthesis on page 9: it should be pulcherriminic acid instead of pulcherrimin.

We have made the corrections.

19. Table S2: please translate the title of the third column into English.

Mistakes has been fixed.

20. For table S3, how were the LOD and LOQ determined? Would be good to add this piece of information to the methods section.

Information for LOD and LOQ calculations has been added in the methods section. Lines 542-545.

21. Lastly, some specific questions and suggestions regarding the NMR spectra:

(a) Please add the integration values onto the H-1 spectrum of molecule (1) on page 5.

We have modified the spectrum to include integrations.

(b) For 3-chloro-2,5-diisobutylpyrazine: the signals at 2.21 and 2.10 ppm do not look like dp to me (also, it's more commonly called quintet rather than pentet). Is the CH proton of the isobutyl group supposed to be a dp at all? Also, I do not agree with the dd assignment for the 0.95 ppm signal. Isn't it corresponding to the two overlapping doublets of the four methyl groups?

We have verified and corrected, when needed, the assignments of each reported compound.

(c) In the spectra of 2,5-dichloro-3,6-diisobutylpyrazine, 2,5-dichloro-3,6-diisobutylpyrazine 1,4-dioxide and 2,5-dihydroxy-3,6-diisobutylpyrazine 1,4-dioxide, the signals labeled as dp should be relabeled as multiplets.

As a general recommendation for the spectra: please double check all splitting pattern assignment, J-values and integrals. Also, it would be more illustrative and easier for the reader if you zoomed in the multiplet signals and included them as insets into the spectra.

We have verified and corrected, when needed, the assignments of each reported compound. We have also inserted insets of zoomed regions of interest on the relevant spectra.

Reviewer #1 (Remarks to the Author):

This is a revised manuscript that I previously reviewed. I am quite pleased with this revised version. The authors have addressed all my main concerns, some with reasonable rebuttal, many with new results from additional experiments.

I only have two very minor points below:

Line 193, The maintenance of ...

Line 294, "As seen in figures 6a and b, WT biofilms showed low ROS levels on MSgg, where PA is secreted in high amounts, and on MS medium, where PA production is minimal."

For WT, the ROS levels are low on MS even though PA production is minimal. This is likely due to a different reason, could the authors briefly explain here ?

Reviewer #2 (Remarks to the Author):

I appreciate that the authors positively took all the feedback and I am satisfied with the new version of their manuscript that they generated.

Response to reviewers

Line 193, The maintenance of ...

The change was made

Line 294, "As seen in figures 6a and b, WT biofilms showed low ROS levels on MSgg, where PA is secreted in high amounts, and on MS medium, where PA production is minimal."

For WT, the ROS levels are low on MS even though PA production minimal. This is likely due to a difference reason, could the authors briefly explain here ?

MS and MSgg media composition differs (see line 99-103), which would explain why there are ROS on MSgg but not on MS. Our main hypothesis is that MSgg induces strong biofilm which causes ROS production via extracellular respiration, while MS does not induce biofilm (see pictures Fig. 1).